# Behavior-dLDS: A decomposed linear dynamical systems model for neural activity partially constrained by behavior

**Eva Yezerets**                                                   *eyezere1@jhu.edu*
*Department of Biomedical Engineering*
*Center for Imaging Science*
*Johns Hopkins University*

**En Yang**                                                          *enyang@unc.edu*
*Department of Biology*
*University of North Carolina*

**Misha B. Ahrens**                                          *ahrensm@janelia.hhmi.org*
*Janelia Research Campus*
*Howard Hughes Medical Institute*

**Adam S. Charles**                                               *adamsc@jhu.edu*
*Department of Biomedical Engineering*
*Center for Imaging Science*
*Johns Hopkins University*

**Reviewed on OpenReview:** *https://openreview.net/forum?id=8p9tP9qLPN*

## Abstract

Brain-wide recordings of large-scale networks of neurons now provide an unprecedented view into how the brain drives behavior. However, brain activity contains both information directly related to behavior as well as the potential for many internal computations. Moreover, observable behavior is executed not only by the brain, but also by the spinal cord and peripheral nervous system. Behavior is a coarse-grained product of neural activity, and we thus take the view that it can be best represented by lower-dimensional latent neural dynamics. Capturing this indirect relationship while disambiguating behavior-generating networks from internal computations running in parallel requires new modeling approaches that can embody the parallel and distributed nature of large-scale neural populations. We thus present behavior-decomposed linear dynamical systems (b-dLDS) to disentangle simultaneously recorded subsystems and identify how the latent neural subsystems relate to behavior. We demonstrate the ability of b-dLDS to decouple behavioral vs. internal computations on controlled, simulated data, showing improvements over a state-of-the-art model that uses behavior to supervise all dynamics based on behavior. We also demonstrate b-dLDS's interpretability benefits on a task-driven RNN dataset featuring a nonlinear relationship between behavior and activations. We then show that b-dLDS can further scale up to tens of thousands of neurons by applying our model to a large-scale recording of a zebrafish hindbrain during the complex positional homeostasis behavior, wherein b-dLDS highlights asymmetry in behavior-related dynamic connectivity networks.

## 1 Introduction

Brains are instrumental in generating and controlling behavior. Such behavior can be fine-grained, e.g., the 3D pose over time of a finger reaching to a screen (Churchland et al., 2012), or more coarse, e.g., an "attack" or "mount" bout within a social interaction (Karigo et al., 2021). However, the neural activity that

ultimately generates behavior is complex and can be nonlinearly or indirectly related to behavior (Urai et al., 2021). These neural processes may not be time-aligned with observable outcomes (Bondy et al., 2025), and are distributed and overlapping or even compositional in time and space in the brain. At the same time, not every behavior is encoded equally in every subnetwork of neurons (International Brain Laboratory et al., 2025; Kashefi et al., 2025; Affan & Scott, 2022; Mu et al., 2020; Wang et al., 2023; Mendoza-Halliday et al., 2024; Chen et al., 2024a). Moreover, these functions may evolve throughout a task (Aoi et al., 2020) and may occur on different timescales (Shi et al., 2025). As the brain is highly parallel and distributed, the same neural activity also encodes many other functions such as hunger and circadian rhythms, task-irrelevant sensory information, higher-order cognitive functions, and emotional responses. As neural recording technologies continue to grow and evolve (Steinmetz et al., 2021; Demas et al., 2021; Yang et al., 2024), the extent of the brain that is captured is larger and more likely to include these many parallel functions that need to be disentangled in order to study the activity underlying the behavior of interest (Urai et al., 2021).

Models that aim to identify meaningful temporal components of neural activity can be behavior-agnostic or behavior-aware. Behavior-agnostic models (Sussillo et al., 2016; Linderman et al., 2016; Mudrik et al., 2024a) use only the neural activity to learn components that can then be correlated *post hoc* to other known information, such as behavior, to "translate" the model variables to address task-relevant questions, such as "Does this model component turn on when the animal enters a reward-seeking state?" However, behavior-aware models can more clearly align learned model components with known behavior variables. Behavior-aware models can also come in two flavors: models that directly relate instantaneous neural activity to instantaneous behavior in a shared latent space (Schneider et al., 2023a; Sani et al., 2021b), or models that connect the latent dynamics to behavior (Geadah et al., 2025). This last class of models is important because it accounts for the fact that the relationship between neural activity and behavior is complex and indirect as described above. Instead, the latent dynamics capture the simultaneous coarse-grained processes the brain "runs," some or all of which may be related to behavior (Karigo & Charles, 2025).

More specific to neural dynamics, existing models live at the two extremes of completely supervised, or completely unsupervised. At the former end of the spectrum are models such as dynamic causal modeling (DCM), and the more recent conditionally linear dynamical systems (CLDS), which consider a linear decomposition of the neural dynamics wherein the coefficients of the decomposition are completely a function of external or experimental variables (e.g., behavior) (Geadah et al., 2025; Friston et al., 2003). At the other end are models such as decomposed linear dynamical systems (dLDS) which, while they do similarly decompose the dynamics into a linear function of basis dynamics, do not tie the dynamics to external variables at all (Mudrik et al., 2024a; Yezerets et al., 2025; Chen et al., 2024b; Mudrik et al., 2025). Instead, the dynamics operators (DOs)—which are learned from the data itself—are assumed to be sparsely represented in the basis, forming a dictionary of dynamics that induces temporal independence of when the different dynamics components are used over time.

Here we present a much needed middle ground between these two approaches: behavior-dLDS (b-dLDS), a model that explicitly ties a small subset of DOs to behavior variables over time (Fig. 1). Since the DOs are the fundamental processes, including behavior, that the brain "runs," and the dynamics coefficients describe which DOs are active at a given time, we relate the behavior traces to the dynamics coefficient traces, where the latent neural dynamics generate behavior and other functions (arrow from $\boldsymbol{c}$ to $\boldsymbol{b}$ in Fig. 1).

b-dLDS is comparable to other works (Geadah et al., 2025; Friston et al., 2003) that use external inputs to condition the model-estimated neural dynamics, based on the assumption that the latent dynamics generate all the neural activity, and the neural activity generates the behavior. The dynamics are thus the best coarse-grained representation of the true underlying processes at play. However, b-dLDS is unique in that it achieves this by requiring the dynamics coefficients to reconstruct the behavior traces via an additional regularization term, rather than conditioning the dynamics directly on the external inputs (e.g., behavior).

Moreover, a Frobenius norm on the mapping from dynamics coefficients to external behavior data in b-dLDS means that it is possible to bias the model to learn that this relationship is sparse. This means that some DOs generate behavior, while the rest of the DOs can generate other brain processes, such as decision-making, homeostasis, etc.

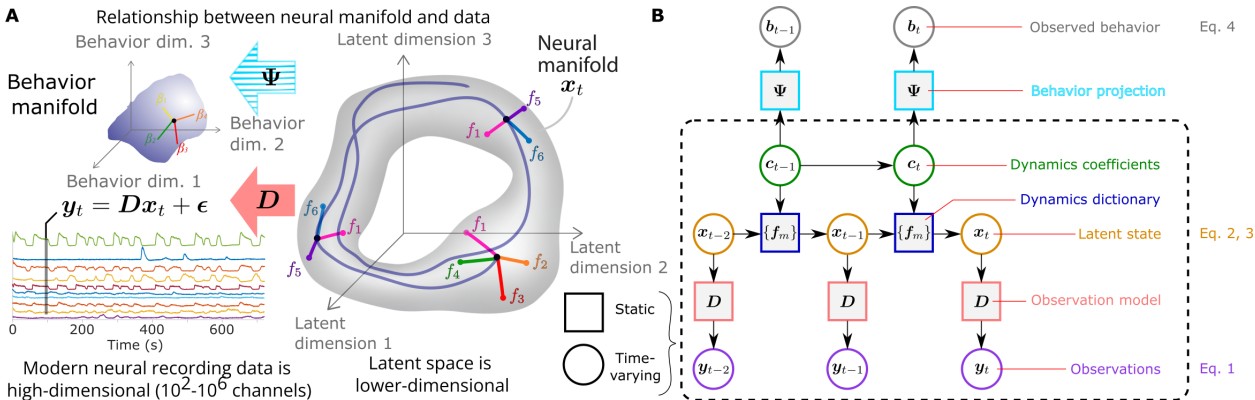

Figure 1: **behavior-dLDS architecture. A:** The latent, lower-dimensional neural manifold represents the processes that generate observable neural activity, mapped to the ambient space via $\boldsymbol{D}$, and behavior, via $\boldsymbol{\Psi}$. **B:** The established dLDS model, inside the dashed lines, is updated here with a mapping from dynamics coefficients $\boldsymbol{c}$ to behavior $\boldsymbol{b}$ via $\boldsymbol{\Psi}$. This contributes another regularization term to the joint inference of latent states $\boldsymbol{x}$ and dynamics coefficients $\boldsymbol{c}$.

In this paper we introduce the b-dLDS model and demonstrate its efficacy in distinguishing between latent neural dynamics that generate behavior vs. other internal computations. We present new optimization terms in the inference and update steps that enable the explicit mapping between neural dynamics and behavior and to encourage sparsity in the dynamics-behavior mapping. We then demonstrate through simulations that b-dLDS can learn a sparse mapping between behavior and dynamics coefficients, and that this sparse mapping makes the dynamics it learns more accurate than those learned by CLDS, a model strongly conditioned on behavior. To test our modeling choice of the dynamics coefficients, we compare using the dynamics coefficients to model behavior against more typical use of latent states to model behavior, in a controlled setting using a task-driven RNN (Driscoll et al., 2024). Finally, we apply b-dLDS to zebrafish hindbrain data, a large-scale dataset of 14,000 neurons, which encodes information in its dynamics related to swimming (the observable behavior) as well as internal computations related to how much to respond to displacement in a current (Yang et al., 2022). b-dLDS can, in this large dataset, identify dynamics that align with behavior and those that do not. Interestingly, these hindbrain networks display a stark asymmetry, an anatomically observed property in zebrafish larvae forebrain (Barth et al., 2005; Roussigné et al., 2012; Dreosti et al., 2014; E. et al., 2014; Lekk et al., 2019; Lekk, 2019; Powell et al., 2024; Powell, 2022; Gobbo et al., 2025) that appears to persist in how the hindbrain is recruited to execute swim bouts.

## 2 Background and Related Work

In neuroscience, a variety of methods have been used to deal with the high dimensionality and complexity of brain-wide or large neural population data and its evolution in time. Dimensionality reduction-only approaches, e.g., principal component analysis (PCA), non-negative matrix factorization (NMF), and independent component analysis (ICA), are unsupervised approaches that enable the remapping of data to dimensions that can be correlated to known features of the experiment or system *post hoc*. More complex approaches, such as Similarity-driven Building-Block Inference using Graphs across States (SiBBlInGS) (Mudrik et al., 2024b) and model-based targeted dimensionality reduction (mTDR) (Aoi et al., 2020), incorporate more supervision from external data, e.g., behavioral task variables. However, matrix factorization-based methods cannot capture complex temporal interactions intrinsic to neural computation.

Generalized linear models (GLMs) fit coefficients that describe explicit relationships between time series data and known external factors (Park et al., 2014; Ashwood et al., 2022). However, GLMs model statistical relationships that are often assumed static, rather than accounting for changing relationships over time, e.g., due to changes in the animal's state or adaptation. Hidden Markov Model GLMs (HMM-GLMs) were thus introduced to learn discrete system states that toggle between different GLM models (Calhoun et al., 2019;

Ashwood et al., 2022; Bolkan et al., 2022). However, HMM-GLMs, as with GLMs, still model the system as a single, monolithic circuit, and do not identify independent subsystems.

Latent dynamical systems model higher-dimensional, potentially noisy data as being generated by a lower-dimensional system of latent variables (Pandarinath et al., 2018a). Dynamical systems can model the relationship between internal states and other external factors, e.g., behavior. Finally, while often assumed to be static, latent dynamics may also evolve over time (nonstationarity) when the system itself is malleable (e.g., adaptation, learning, or representational drift in neuroscience).

**Dynamical systems in neuroscience:** In neuroscience, dynamical systems models have included both linear and nonlinear variants. Most such models (including linear (Sani et al., 2021b; Golub et al., 2013; Churchland et al., 2012); nonlinear (Pandarinath et al., 2018b; Keshtkaran & Pandarinath, 2019; Sussillo et al., 2015; Sani et al., 2021c; Kleinman et al., 2021; Kudryashova et al., 2025); and switching (Ghahramani & Hinton, 1996; Murphy, 1998; Fox et al., 2008; Linderman et al., 2017; Nassar et al., 2018) variants) treat all the data as coming from a single system, similar to the GLM. Mathematically, if the $N$-dimensional system state at time $t$ is $\boldsymbol{x}_t \in \mathbb{R}^N$, then $\boldsymbol{x}_t \approx g(\boldsymbol{x}_{t-1})$ for some function $g : \mathbb{R}^N \to \mathbb{R}^N$ that may be nonlinear or even time-varying.

More recently, the need to model larger-scale recordings that contain multiple biological subsystems has required new models that can identify these meaningful subsystems by decomposing the dynamics into combinations of systems (Friston et al., 2003; Mudrik et al., 2024a; Yezerets et al., 2025; Chen et al., 2024b; Mudrik et al., 2025; Geadah et al., 2025). In these models, the dynamics are linear at each time point, i.e., $\boldsymbol{x}_t \approx \boldsymbol{F}_t \boldsymbol{x}_{t-1}$ for a dynamics transition matrix $\boldsymbol{F}_t \in \mathbb{R}^{N \times N}$. Each $\boldsymbol{F}_t$ can be expanded into a basis as $\boldsymbol{F}_t = \sum_{m=1}^M \boldsymbol{f}_m c_{mt}$, where the basis elements $\boldsymbol{f}_m \in \mathbb{R}^{N \times N}$ do not change over time, however the expansion coefficients (called dynamics coefficients) $c_{mt}$ do. By varying the dynamics coefficients, this model class can capture nonlinear and nonstationary behavior, providing a flexible descriptive framework while maintaining a local sense of interpretability through the linear formulation at each time point.

Two classes of such decomposed linear dynamics models have emerged. On one end is the work in dynamic causal modeling (DCM) (Friston et al., 2003) and conditionally linear dynamical systems (CLDS) (Geadah et al., 2025), wherein the dynamics coefficients $c_{mt}$ are functions of external or experimental variables (e.g., behavioral quantities). In this case the basis for the dynamics is typically fixed, and it is the mapping from behavioral variables $\boldsymbol{b}_t$ to dynamics coefficients $\boldsymbol{c}_t$ that is learned. On the other end is the completely unsupervised decomposed linear dynamical systems (dLDS) model (Mudrik et al., 2024a; Yezerets et al., 2025) and extensions thereof (Mudrik et al., 2025; Chen et al., 2024b), which take a completely unsupervised approach to learning the dynamics basis (or dictionary) from data. In this setting the $c_{mt}$ are assumed unknown latent variables with sparse statistics. Sparsity in this case encourages the learned basis to be statistically independent in terms of when they are active, i.e., they truly represent "different" processes. This model can be optimized via an expectation-maximization (EM) procedure, which results in a dictionary-learning-like iterative algorithm that alternates between fitting the dynamics coefficients and the dynamics basis elements.

**Other models of brain and behavior:** Building on the advances in dynamical systems, further work has sought to devise joint models of neural activity and behavior. In the Preferential Subspace Identification (PSID) algorithm (Sani et al., 2021b), behavior and neural activity come from a shared latent state space, which is split into neural- and behavior-related subspaces. In other words, the underlying state generating neural activity and behavior is assumed to be the same at each time point; however, this may not necessarily be true if there is a hierarchical relationship between the latent dynamical systems, the behaviors they represent, and neural activity they generate. It is possible that only a subset of the dynamics represent behavior-related processes, while another subset relate to internal cognitive variables not observed in the external input. Moreover, in the underdetermined case where there are more neurons than time points, PSID cannot compute the matrix inverse. In order to model such data, one is obliged first to perform dimensionality reduction such as PCA, which runs counter to the aims of b-dLDS.

We note here a model, Consistent EmBeddings of high-dimensional Recordings using Auxiliary variables (CEBRA), that jointly embeds neural activity and behavior into a shared latent space and models this over

time (Schneider et al., 2023a). However, CEBRA is not a dynamical systems model, nor is it linear, thus lacking the interpretability offered by PSID, CLDS, and b-dLDS.

In comparison, decomposed linear dynamical systems (dLDS) models, including b-dLDS, which we introduce in this work, generate a decomposition of dynamical systems, allowing multiple systems to be co-active at the same time. dLDS-family models are, to our knowledge, also unique in the space of dynamical systems models for their application to large-scale neural data, capturing up to thousands of neurons (results and prior work in Mudrik et al. (2025)). This capacity is possible because dLDS-family models do not require the calculation of high-dimensional covariance matrices and can leverage efficient sparse-state filtering algorithms (Charles et al., 2016).

## 3 Behavior-dLDS Model

We begin by modeling the observed data $\boldsymbol{y}_t \in \mathbb{R}^P$, in this case neural data, as representable by a linear latent variable model $\boldsymbol{x}_t \in \mathbb{R}^N$ at each time point $t$. We assume that $\boldsymbol{x}_t$ is lower-dimensional ($N < P$) and can describe the data through the linear generative model

$$\boldsymbol{y}_t = \boldsymbol{D}\boldsymbol{x}_t + \epsilon_y, \tag{1}$$

where $\boldsymbol{D} \in \mathbb{R}^{P \times N}$ is the observation matrix, and $\epsilon_y$ represents independent observation noise, which we model as Gaussian. We next model the dynamics in the latent space $\boldsymbol{x}_t$ with a time-varying linear dynamical system

$$\boldsymbol{x}_t = \boldsymbol{F}_t\boldsymbol{x}_{t-1} + \epsilon_x, \tag{2}$$

where, as in dLDS, we model $\boldsymbol{F}_t$ as a function of a linear combination of dynamics operators (DOs) $\boldsymbol{f}_m$, weighted by their corresponding dynamics coefficients at each time point $c_{mt}$:

$$\boldsymbol{F}_t = \sum_{m=1}^{M} \boldsymbol{f}_m c_{mt}, \tag{3}$$

and $\epsilon_x$ is the dynamics innovations error, or the error in modeling the dynamics, which we also model as Gaussian.

The latent states $\boldsymbol{x}_t$ are a lower-dimensional representation of the instantaneous neural activity, where each dimension represents a group of neurons. We focus our analysis on the latent dynamics $\boldsymbol{F}_t$, rather than on the latent states $\boldsymbol{x}_t$, because the dynamics $\boldsymbol{F}_t$ represent the ways that these different groups of neurons can act on each other (the influence of neuron group $j$ at time $t-1$ on neuron group $i$ at time $t$, i.e., the space of "processes" that the brain can run), whereas the latent states $\boldsymbol{x}_t$ provide a more instantaneous snapshot of which groups of neurons are active.

To complement the dynamics model, we introduce the main innovation in b-dLDS: We consider the observations of behavioral variables as the time series $\boldsymbol{b}_t \in \mathbb{R}^K$ as a function of the *change in state* rather than the state itself. Specifically, we account for the fact that the current neural state $\boldsymbol{x}_t$ might fluctuate too quickly to capture the timescale of behaviors, especially those in the action domain, rather than the instantaneous "position" domain, by modeling the behavior as being generated by the dynamics coefficients $\boldsymbol{c}_t = [c_{1t}, ..., c_{Mt}]^T \in \mathbb{R}^M$. Mathematically, we model the behavior through a complementary linear generative model

$$\boldsymbol{b}_t = \boldsymbol{\Psi}\boldsymbol{c}_t + \boldsymbol{\epsilon}_b. \tag{4}$$

where $\boldsymbol{\Psi} \in \mathbb{R}^{K \times M}$ projects the current dynamics into the behavior and $\boldsymbol{\epsilon}_b$ is *i.i.d.* Gaussian measurement noise.

While some of the dynamics observed in brain-wide data are assumed to be related to the observed behavior $\boldsymbol{b}_t$, large-scale recordings are likely to capture multiple parallel networks that include activity related to either behavioral quantities unknown to the experimenter or internal computations that unfold in alongside behavior-related activity. To model this effect, we assume that $\boldsymbol{\Psi}$ is *incomplete* in that only a portion of

the $\boldsymbol{c}_t$ variables (say for $m \in \Gamma \subseteq [1,..,M]$ with $|\Gamma| = M' < M$) actually relate to $\boldsymbol{b}_t$. This means that while the model has appropriate information to orient $c_{mt}$ for $m \in \Gamma$, $c_{mt}$ for $m$ in the complement to $\Gamma$, $\Gamma^C$, require additional regularization. This challenge can be cast as a type of missing regressor problem, which can be mended through appropriate regularization over $\boldsymbol{c}_t$ (Ibrahim et al., 2005; Gauthier et al., 2022). Specifically, we introduce regularization such that the minimum number of elements of $\boldsymbol{c}_t$ is nonzero at each $t$. This parsimoniousness is achieved through sparsity regularization via a Laplacian prior over $\boldsymbol{c}_t$: $p(\boldsymbol{c}_t) = (\lambda/2)^M e^{-\lambda \|\boldsymbol{c}_t\|_1}$.

## 4 Model Inference

Inference under this model estimates the model parameters $\boldsymbol{D}$, $\boldsymbol{\Psi}$, and $\boldsymbol{f}_m$ for $m \in [1,...,M]$, which we combine into the parameter variable $\theta = \{\boldsymbol{D}, \boldsymbol{\Psi}, \{\boldsymbol{f}_m\}_{m=1,...,M}\}$. Ideally we would like to take a Maximum Likelihood (ML) approach to find

$$\widehat{\theta} = \arg \min_{\theta} - \log p(\{\boldsymbol{y}_t\}_{t=1,...,T} | \theta). \tag{5}$$

However, $\boldsymbol{c}_t$ represents missing information, i.e., we cannot compute the likelihood $p(\{\boldsymbol{y}_t\}_{t=1,...,T} | \theta)$ directly. Instead, we perform expectation-maximization, an iterative procedure for estimating model parameters and updating their inferred coefficients accordingly until convergence. Specifically, as with other dictionary learning algorithms (Olshausen & Field, 1996; Geadah et al., 2024; Mudrik et al., 2024a), the EM algorithm takes a Dirac-delta approximation to the posterior, resulting in a MAP estimation over the latent variables $\boldsymbol{x}_t$, and $\boldsymbol{c}_t$ conditioned on $\theta$, and then an optimization over $\theta$.

**E-step: inferring $\boldsymbol{c}_t$ and $\boldsymbol{x}_t$.** Based on an initial set of parameters $\theta$ (or updated $\theta$ in subsequent iterations), the E-step optimizes the most probable parameters for $\boldsymbol{c}_t$ and $\boldsymbol{x}_t$ given the data and $\theta$:

$$\arg \max_{\boldsymbol{c},\boldsymbol{x}} p\left(\{\boldsymbol{c}_t\}_{t \in [1,T]}, \{\boldsymbol{x}_t\}_{t \in [1,T]} | \{\boldsymbol{y}_t\}_{t \in [1,T]}, \theta\right). \tag{6}$$

This inference problem can be extremely high-dimensional, especially for large quantities of data and long time series. For efficiency we subsample the data to perform a batch update. Furthermore, we factor the estimation problem over time steps and use dynamic filtering (BPDN-DF) (Charles et al., 2016) to iteratively estimate for each time $t$ the latent states $\boldsymbol{x}_t$ and the dynamics coefficients $\boldsymbol{c}_t$ given the estimates at the previous time step $\widehat{\boldsymbol{x}}_{t-1}$ and $\widehat{\boldsymbol{c}}_{t-1}$:

$$\arg \min_{\boldsymbol{x}_t, \boldsymbol{c}_t} \left[ \|\boldsymbol{y_t} - \boldsymbol{D}\boldsymbol{x}_t\|_2^2 + \lambda_0 \left\|\boldsymbol{x}_t - \widetilde{\boldsymbol{F}}_t \boldsymbol{c}_t\right\|_2^2 + \lambda_1 \|\boldsymbol{x}_t\|_1 + \lambda_2 \|\boldsymbol{c}_t\|_1 + \lambda_3 \|\boldsymbol{c}_t - \widehat{\boldsymbol{c}}_{t-1}\|_2^2 + \lambda_4 \|\boldsymbol{b}_t - \boldsymbol{\Psi}\boldsymbol{c}_t\|_2^2 \right], \tag{7}$$

where the $\widetilde{\boldsymbol{F}}_t = [\boldsymbol{f}_1 \widehat{\boldsymbol{x}}_{t-1}, \boldsymbol{f}_2 \widehat{\boldsymbol{x}}_{t-1}, ..., \boldsymbol{f}_M \widehat{\boldsymbol{x}}_{t-1}]$, i.e., the $m^{th}$ column of $\widetilde{\boldsymbol{F}}_t$ is the past estimate $\widehat{\boldsymbol{x}}_{t-1}$ projected through the $m^{th}$ operator $\boldsymbol{f}_m$. The BPDN-DF procedure is known to converge (Charles & Rozell, 2014; Charles et al., 2016), providing an accurate representation of the latent states via a single pass over time. Note that here, the regularization $\lambda_4$ over the behavior reconstruction term is the key technical innovation in b-dLDS.

**M-step: Updating $\theta$.** The parameter set $\theta$ consists of $M + 2$ matrices ($M$ DOs, the mapping from $\boldsymbol{y}$ to $\boldsymbol{x}$, and the mapping from $\boldsymbol{c}$ to $\boldsymbol{b}$). As the datasets are large and the E-step is only run on a random subset of the data, rather than completely optimize $\theta$ between each E-step, i.e.,

$$\widehat{\theta} = \arg \max_{\theta} p(\{\boldsymbol{y}_t\}_{t \in T} | \{\boldsymbol{c}_t\}_{t \in T}, \{\boldsymbol{x}_t\}_{t \in T}, \theta), \tag{8}$$

we instead take a (noisy) descent step over the approximate negative log-likelihood given the inferred values of $\boldsymbol{c}_t$, $\boldsymbol{x}_t$:

$$\mathcal{L} = \|\boldsymbol{y_t} - \boldsymbol{D}\boldsymbol{x}_t\|_2^2 + \lambda_0 \left\|\boldsymbol{x}_t - \sum_{m=1}^{M} \boldsymbol{f}_l c_{lt} \boldsymbol{x}_{t-1}\right\|_2^2. \tag{9}$$

The updates for $\boldsymbol{D}$, $\boldsymbol{F}$, and $\boldsymbol{\Psi}$ are then computed via an effective stochastic projected gradient descent. For $\boldsymbol{D}$ we update the estimate as

$$
\begin{aligned}
\widehat{\boldsymbol{D}} \;&\leftarrow\; \Pi_{C_D}\left(\boldsymbol{D} - \eta_D \nabla_{\boldsymbol{D}} \sum_{t=1}^{T} \|\boldsymbol{y}_t - \boldsymbol{D}\boldsymbol{x}_t\|_2^2\right) \\
&=\; \Pi_{C_D}\left(\boldsymbol{D} + \eta_D \sum_{t=1}^{T}(\boldsymbol{y}_t - \boldsymbol{D}\boldsymbol{x}_t)\boldsymbol{x}_t^T\right),
\end{aligned}
\tag{10}
$$

where $\Pi_{C_D}$ is an optional projection onto any constraint sets (e.g., non-negativity). Similarly, the gradient over each $\boldsymbol{f}_m$ can be computed as

$$
\widehat{\boldsymbol{f}}_m = \Pi_{C_{f_m}}\left(\widehat{\boldsymbol{f}}_m + \eta_f \sum_{t=2}^{T}\left(\boldsymbol{c}_{mt}\left(\boldsymbol{x}_t - \widetilde{\boldsymbol{F}}_t \boldsymbol{c}_t\right)\boldsymbol{x}_{t-1}^T\right)\right),
\tag{11}
$$

where $\Pi_{C_f}$ is an optional constraint set over $\boldsymbol{f}$ and $\eta_f$ is the step size. Finally, $\boldsymbol{\Psi}$, the new mapping between dynamics coefficients and behavior introduced in b-dLDS, is updated similarly as

$$
\widehat{\boldsymbol{\Psi}} \leftarrow \Pi_{C_\Psi}\left(\boldsymbol{\Psi} - \widetilde{\eta}_\Psi \nabla_\Psi \sum_{t=1}^{T}(\boldsymbol{b}_t - \boldsymbol{\Psi}\boldsymbol{c}_t)^2\right),
\tag{12}
$$

for constraint set projection $\Pi_{C_\Psi}$ and step size $\widetilde{\eta}_\Psi$. For $\boldsymbol{\Psi}$ we find that rather than setting the step size to be constant (or through a data-agnostic schedule), it is more efficient to rescale a base step size $\eta_\Psi$ based on the norm of the dynamics coefficients as $\widetilde{\eta}_\Psi = \eta_\Psi / \sum_{t=1}^{T}(\boldsymbol{c}_t)^2$. This rescaling makes sure that the step size is appropriately sized in cases when there are many behavior outputs but only one or a few dynamics coefficients generating them (which we term the "sparse suspected" option for $\widehat{\boldsymbol{\Psi}}$ step size rescaling).

We also use the Frobenius norm option to allow columns of $\boldsymbol{\Psi}$ with very small values to be regularized to 0, which is especially useful in cases where there are many behavior outputs but only one or a few dynamics coefficients generating them, as in the simulations presented below:

$$
\begin{aligned}
\widehat{\boldsymbol{\Psi}} \;&\leftarrow\; \Pi_{C_\Psi}\left(\boldsymbol{\Psi} - \tilde{\eta}_\Psi \nabla_\Psi\left(\sum_{t=1}^{T}(\boldsymbol{b}_t - \boldsymbol{\Psi}\boldsymbol{c}_t)^2 + \lambda_\Psi\|\boldsymbol{\Psi}\|_F^2\right)\right) \\
&=\; \Pi_{C_\Psi}\left(\boldsymbol{\Psi} + \tilde{\eta}_\Psi\left(\sum_{t=1}^{T}(\boldsymbol{b}_t - \boldsymbol{\Psi}\boldsymbol{c}_t)\boldsymbol{c}_t^T - 2\lambda_\Psi\boldsymbol{\Psi}\right)\right),
\end{aligned}
\tag{13}
$$

where $\lambda_\Psi$ is a regularization parameter on the Frobenius norm of $\boldsymbol{\Psi}$. By contrast, in the neural data, we expect that multiple dynamics coefficients participate in generating the one- or few-dimensional observed behavior, so the Frobenius norm and "sparse suspected" options are not used. Supplementary Figure 5 shows an ablation study on these two $\boldsymbol{\Psi}$-related settings on simulated data.

## 5 Results

**b-dLDS learns which subset of dynamics generated behavior in simulation.** Mudrik et al. (2024a) showed that dLDS could recover true coefficients and operators from a simulation with two sets of independent systems. b-dLDS can do the same, but it can *also* recover the relationship between simulated behavior and dynamics coefficients.

We present four cases: a simplified, sparse case, without added noise, where all of the behavior is generated from one dynamics coefficient and thus tied to only one group of simulated "latent neural dimensions" (Fig. 2); a more complex but still sparse case, where two latent neural dimensions account for the behavior (Supp.Fig. 1); a complex case, where all dynamics coefficients contribute to generating the behavior (Supp. Fig. 2); and a case with one dynamics coefficient used to generate behavior, with added noise (Supp.Fig. 3; see Appendix Section "Simulation of two independent systems and behavior").

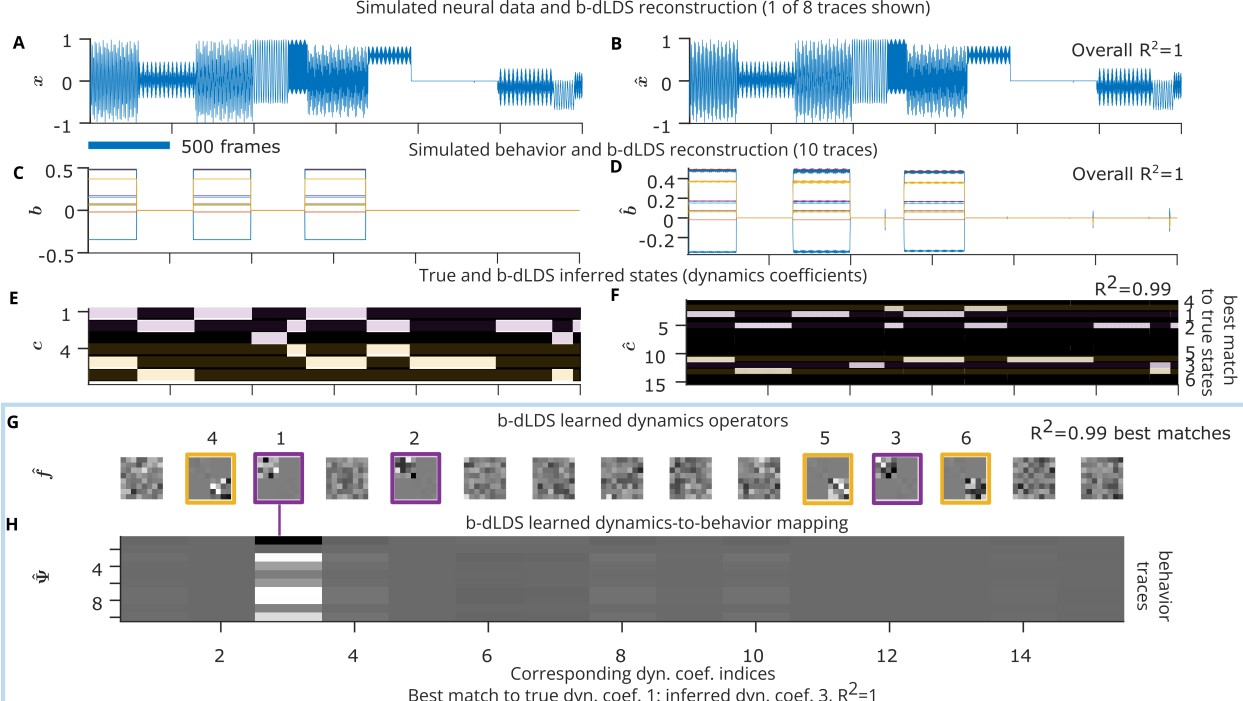

Figure 2: **Simulated system composed of two independent sub-systems with 10 behavior traces.** Example trial (of 50 randomly sampled trials); $R^2$ calculated over all 50 samples. **A,B:** "Neural" ground truth and reconstruction by b-dLDS. Signal is generated from a set of 6 ground truth dynamics operators and corresponding dynamics coefficients, with $\boldsymbol{D} = \boldsymbol{I}$. **C,D:** Simulated behavior and reconstruction from b-dLDS, where behavior is simulated from the ground truth dynamics coefficients. **E:** True dynamics coefficients used to generate the data. **F:** Learned dynamics coefficients (absolute value). **G:** Learned dynamics operators (1-15). **H:** Learned $\boldsymbol{\Psi}$, where each column corresponds to a dynamics operator and dynamics coefficient. Only column 3 is nonzero, and it perfectly reconstructs column 1 of the ground truth $\boldsymbol{\Psi}$. This is the correct column to reconstruct because the first ground truth dynamics coefficient was used to generate the simulated behavior traces.

These four cases correspond to three simulations: one where all 10 simulated behavior traces are random multiples of the first simulated dynamics coefficient; one with two overlapping groups of simulated behavior traces, where traces 1-8 are tied via $\boldsymbol{\Psi}$ to the first dynamics coefficient and 3-10 are tied to the second dynamics coefficient; and one where all 10 simulated behavior traces are mapped via a simulated $\boldsymbol{\Psi}$ that is a completely random linear combination of all 6 simulated dynamics coefficients. In the first scenario, one column of the learned $\boldsymbol{\Psi}$ should match the simulated $\boldsymbol{\Psi}$ with a high $R^2$. In the second scenario, two columns of $\boldsymbol{\Psi}$ should match. In the third, six columns should match. Note that in each simulation, there are two independent systems, either regulated by upper-left corner (purple) or lower-right corner (yellow) ground truth DOs (corresponding to ground truth states 1-3 or 4-6), with no overlap between the systems. Moreover, these simulations are further simplified by the fact that the latent states are the observed states, i.e., $\boldsymbol{D} = \boldsymbol{I}$. In Figure 2 and Supplementary Figures 1, 2, and 3, we show that behavior-dLDS learned which dynamics coefficient or coefficients were used to generate simulated behavior through the variable $\boldsymbol{\Psi}$ (Appendix Section "Experiment Details"). We chose 15 as the number of dynamics operators for b-dLDS based on >2x the number of ground truth dynamics operators used to generate the data and approximately 2x the number of latent states.

We further compared to the most similar and recent method, CLDS (Geadah et al., 2025), which can learn dynamics conditioned on external input (Fig. 3). We tested CLDS on the simulation where only 1 ground truth dynamics coefficient was used to generate only 1 behavior trace. All of the other dynamics coefficients were independent of the behavior. In addition, we included gradations in the values of the true dynamics

coefficients in the "on" state (rather than just 1 or 0). We show that while b-dLDS was able to reconstruct the ground truth dynamics (linear combination of $\boldsymbol{f}$ and $\boldsymbol{c}_t$), CLDS, whose dynamics matrix is represented by the variable $\boldsymbol{A}_t$, could not because CLDS is so strongly conditioned on the one behavior it sees. This simulation is comparable to a large-scale recording where one region's activity is strongly correlated with the behavior, but the other regions in the recording are not. This simulation was designed to distinguish b-dLDS from CLDS on the basis of the conceptual differences between the two models, where b-dLDS permits the decomposition of simultaneously active, independent systems, while CLDS does not. (This is not intended to be a comprehensive comparison of their reconstruction performance.)

We then compared CLDS to b-dLDS on this same simulation across multiple settings, with 10 random seeds in each. The full set of ground truth dynamics coefficients is 6, so that is the maximum possible number of nonzero columns of $\boldsymbol{\Psi}$. We created simulated datasets with 1 to 6 nonzero columns of $\boldsymbol{\Psi}$ linearly combined to generate 1 simulated behavior. b-dLDS achieved lower relative mean squared error (relative MSE) on the reconstructed dynamics matrix than CLDS did, regardless of whether all or only a subset of the true dynamics generated the behavior (Fig. 3K). CLDS was once again so strongly conditioned on the behavior that it could not learn the intrinsic dynamics, even when all of the intrinsic dynamics were linearly combined to generate the behavior.

Finally, we compared b-dLDS, which ties behavior to dynamics, to PSID, which instead ties behavior to the latent state (Fig. 3H,J), using the same type of simulation as above for CLDS, for similar conceptual reasons. Because PSID uses a shared neural and behavior latent state, it cannot handle this simulated case, where the neural activity is rapidly oscillatory, while the latent neural dynamics and the behavior are scaled versions of on-off states. PSID is overly biased by the neural activity in this case, correctly registering the time of switches in dynamics, but incorrectly predicting oscillatory behavior outputs instead of smooth states.

**b-dLDS identifies a nonlinear mapping between RNN activations and input/output behavior.** To complement the two-system simulation, we further tested b-dLDS on an RNN trained to perform a delay-to-response task from Driscoll et al. (2024). Specifically, this example was selected to test, in a controlled setting, the choice of having the behavioral variables modeled as a function of the dynamics coefficients $\boldsymbol{c}$ against the more traditional choice of having the behavior be a function of the latent space $\boldsymbol{x}$. To most directly perform this comparison we implemented b-dLDS-x, a variation of b-dLDS where the behavior is mapped to the latent states, i.e., $\boldsymbol{b}_t = \boldsymbol{\Psi}\boldsymbol{x}_t$ For the task-driven RNN, we selected the RNN that performed the delayanti task as per Driscoll et al. (2024). This dataset is interesting because the input and output traces have distinct on and off states, whereas the RNN activations rise and fall continuously around transition points. The delayanti task also requires the activations to process inputs, hold memory, and produce outputs in distinct time periods. Thus there is a nonlinear relationship between the activations and behavior, and the activations can be reasonably supposed to have compositional dynamics in order to process, remember, and respond to the task. We note that in this dataset there are 20 input traces and 3 output traces, all of which we input to the model as "behavior" traces. We chose 16 as the number of dynamics operators for b-dLDS and b-dLDS-x based on 2x the number of principal components PCA would require to achieve >90% variance explained (see Appendix Section "Task-driven RNN" for more details on model parameter settings).

To fit the regularization parameters, we used Bayesian optimization (BayesOpt - MATLAB) (30 iterations, multiple parameters co-optimized), b-dLDS and b-dLDS-x were tuned to maximize behavior reconstruction. With the optimized parameters, we found that both b-dLDS and b-dLDS-x were able to achieve comparable reconstruction fidelity of the behavior in the task-driven RNN (Fig. 4, Table 1), with the b-dLDS reconstruction $R^2$ slightly higher than b-dLDS-x: b-dLDS achieved a behavior reconstruction $R^2$ of 0.99, while b-dLDS-x achieved a behavior reconstruction $R^2$ of 0.95 (Fig. 4A,B,D).

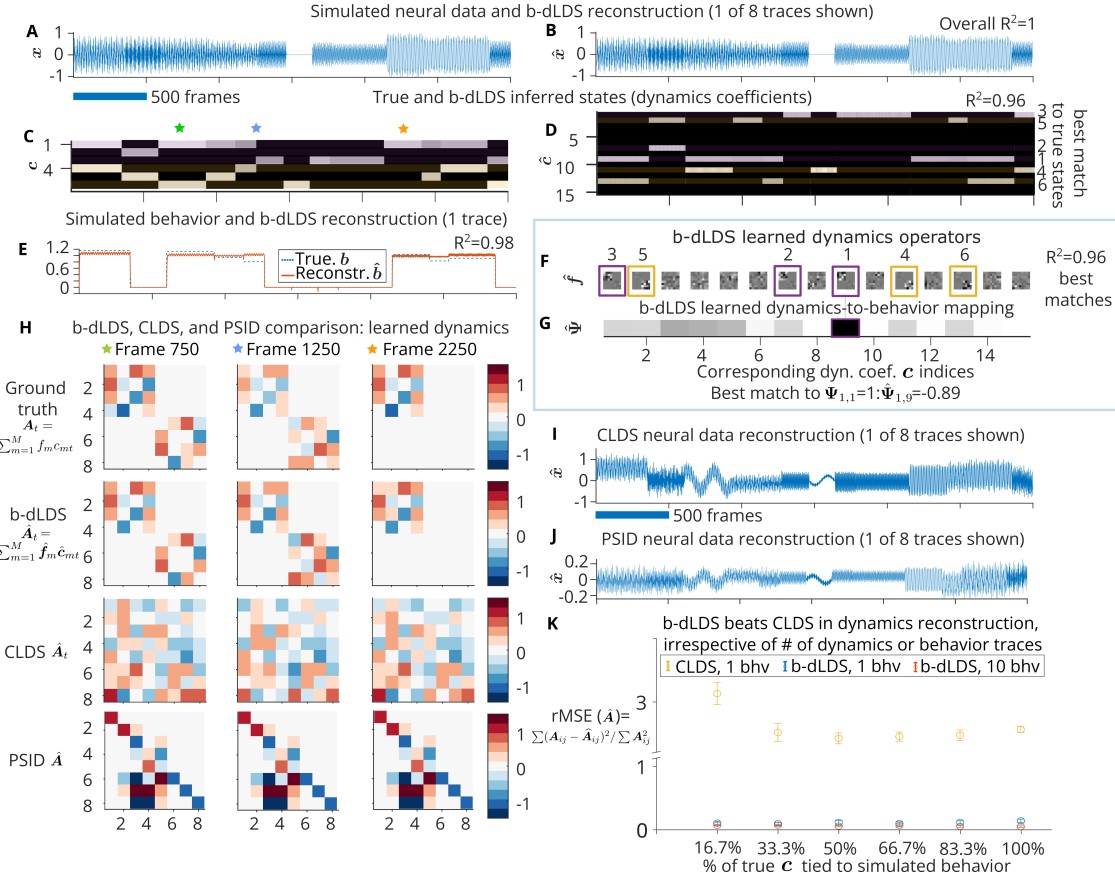

Figure 3: **b-dLDS vs. CLDS and PSID.** CLDS learns only behavior-conditioned dynamics, while b-dLDS learns both behavior-generating and intrinsic ones. PSID learns only a shared latent neural-behavioral space, while b-dLDS learns a hierarchical relationship between latent neural states, latent dynamics, and behavior. (Note: CLDS requires an 80/20 train/test split for optimization, so all models (CLDS, b-dLDS, and PSID) were only trained/learned on 40 out of 50 simulated trials (where CLDS then saw the remaining 10 trials for testing). Results shown here are on the same 40 "train" trials across models.) **A-G:** b-dLDS, 2 independent systems simulation, 1 trial out of 40 shown, $D = I$, with simulated behavior generated as a linear combination of ground truth dynamics coefficients. $R^2$ calculated across 40 trials. **A, B:** 1 trace out of 8 shown, data and reconstruction. **E:** True dynamics coefficients $c$. Stars match selected time points in **H**. Note that at the first two time points, one top-left (purple) dynamics operator and one bottom-right (yellow) dynamics operator are active, while at the third time point, only a top-left dynamics operator is active. **D:** b-dLDS-inferred $c$. The b-dLDS model is initialized with more than enough dynamics operators (15 learned vs. 6 ground truth operators). Noisy dynamics operators in **F** correspond to unused (black) dynamics coefficients in **D**. **E:** Behavior and reconstruction (1 trace). **G:** $\Psi$ is a row vector with one dynamics coefficient mapping to one behavior (true value: 1, learned value: 0.8765, other true values 0, learned near 0). **H:** CLDS learns an 8x8 dynamics matrix $\widehat{A}_t$, while b-dLDS learns $\widehat{f}$ and $\widehat{c}$, which can be combined via $\sum_{m=1}^{M} \widehat{f}_m \widehat{c}_{mt}$ to create an equivalent matrix. We show 3 example time points here as described above. b-dLDS learns the block-diagonal dynamics matrices, while CLDS does not. The full comparison across time is shown in Figure 4. PSID learns a dynamics matrix $\widehat{A}$ that describes the relationship between latent states. **I:** CLDS neural data reconstruction (1st trace). **J:** PSID neural data reconstruction (1st trace). **K:** b-dLDS achieves lower relative mean squared error than CLDS on dynamics reconstruction, regardless of the number of dynamics or the number of behaviors. CLDS used more than 1 TB of RAM when we tried to run it on 10 behavior traces.

Table 1: **b-dLDS vs. b-dLDS-x $\Psi$ sparsity and behavior reconstruction for the task-driven RNN (Driscoll et al., 2024) across levels of regularization on the Frobenius norm of $\Psi$.** For b-dLDS, we start with 5 as this is the setting used in the model shown in Figure 4B. For b-dLDS-x, we start with 0.9644 as this is the BayesOpt-identified setting used in the model shown, and then compare to other stronger regularization settings.

| Model | Iterations | $\lambda_\Psi$ | $L_1/L_2$ norm | $R_b^2$ |
|---|---|---|---|---|
| b-dLDS | 500 | 5 | 2.6 | 0.99 |
| | | 50 | 0.66 | 0.99 |
| | | 100 | 0.39 | 0.99 |
| | | 200 | 0.07 | 0.96 |
| b-dLDS-x | 500 | 0.9644 | 3.88 | 0.95 |
| | | 5 | 4.44 | 0.96 |
| | | 50 | 7.85 | 0.93 |
| | | 100 | 7.33 | 0.91 |
| | | 200 | 8.18 | 0.88 |

Despite the fact that b-dLDS-x was able to get close to the reconstruction performance of b-dLDS, the nonlinear relationship between activations and behavior forced b-dLDS-x to learn a much more complex mapping in order to approximate the behavior than b-dLDS. This effect instantiated in two primary qualities of the fit model. First off, b-dLDS was able to learn a mapping where a very small number of $\widehat{c}$ could well represent all of the behaviors, whereas b-dLDS-x consistently required a large number of latent variables $x$ to represent each behavior. This dense representation persisted even when the sparsity of $\widehat{\Psi}$ was strongly promoted via regularization (Fig. 4F, Table 1). When comparing the underlying dynamics coefficient traces $\widehat{c}_t$ from b-dLDS vs. b-dLDS-x (Fig. 4B vs. D), we also note that b-dLDS-x has to learn a more complex blend of multiple systems turning on and off in quick succession during the behavior on-off events. b-dLDS, however, could fit the data even better with a simpler set of linear dynamics. This implies that (1) the underlying activations of the RNN *can* be well represented by epochs of simple combination of linear dynamical systems but that the behavior is more related to the nature of the internal activation dynamics than the exact units that are on or off. To state another way: the interactions are more concentrated, concise representations of the inputs and outputs than the activity levels of individual units, validating our modeling choice of $b_t \approx \Psi c_t$ over $b_t \approx \Psi x_t$.

**b-dLDS highlights behavior-related dynamic connectivity and asymmetry across the zebrafish hindbrain.** Given our synthetic data examples, we next test the ability of b-dLDS to learn dynamics representations in large-scale neural data. Specifically, we focus on the zebrafish model organism, where advances in optical imaging have enabled simultaneous recording of large areas (up to the full brain) of zebrafish larvae. Specifically, we applied b-dLDS to the zebrafish hindbrain recordings from Yang et al. (2022). In this experiment, the fish performed a complex sensorimotor navigation-related behavior called positional homeostasis. This behavior enables the fish to respond to changes in a current and not get swept away, and it requires the fish to integrate its velocity and displacement and calibrate its swimming response accordingly. This dataset features one-dimensional motor information as the behavior trace, and approximately 14,000 neurons recorded over 12 minutes in a single zebrafish hindbrain, tested in a closed loop virtual reality system (for more on this data, preprocessing, and model parameter settings, see Appendix Section "Zebrafish Data"). We used b-dLDS to plot dynamic connectivity maps corresponding to DOs whose dynamics coefficients have the strongest vs. weakest coefficients in the learned $\widehat{\Psi}$ (Fig. 5 vs. Supp. Fig. 6). In a second version of the model, we used both the motor signal and trial types as a four-dimensional "behavior" input to the model (Supp. Figs. 7, 8, 9, and 10; see Appendix Section "Trial types"). We chose 25 dynamics operators since we observed that, while 50 latent states were needed to reconstruct the neural data, even half as many dynamics operators could be used without loss of reconstruction performance, providing a sparser and more interpretable model.

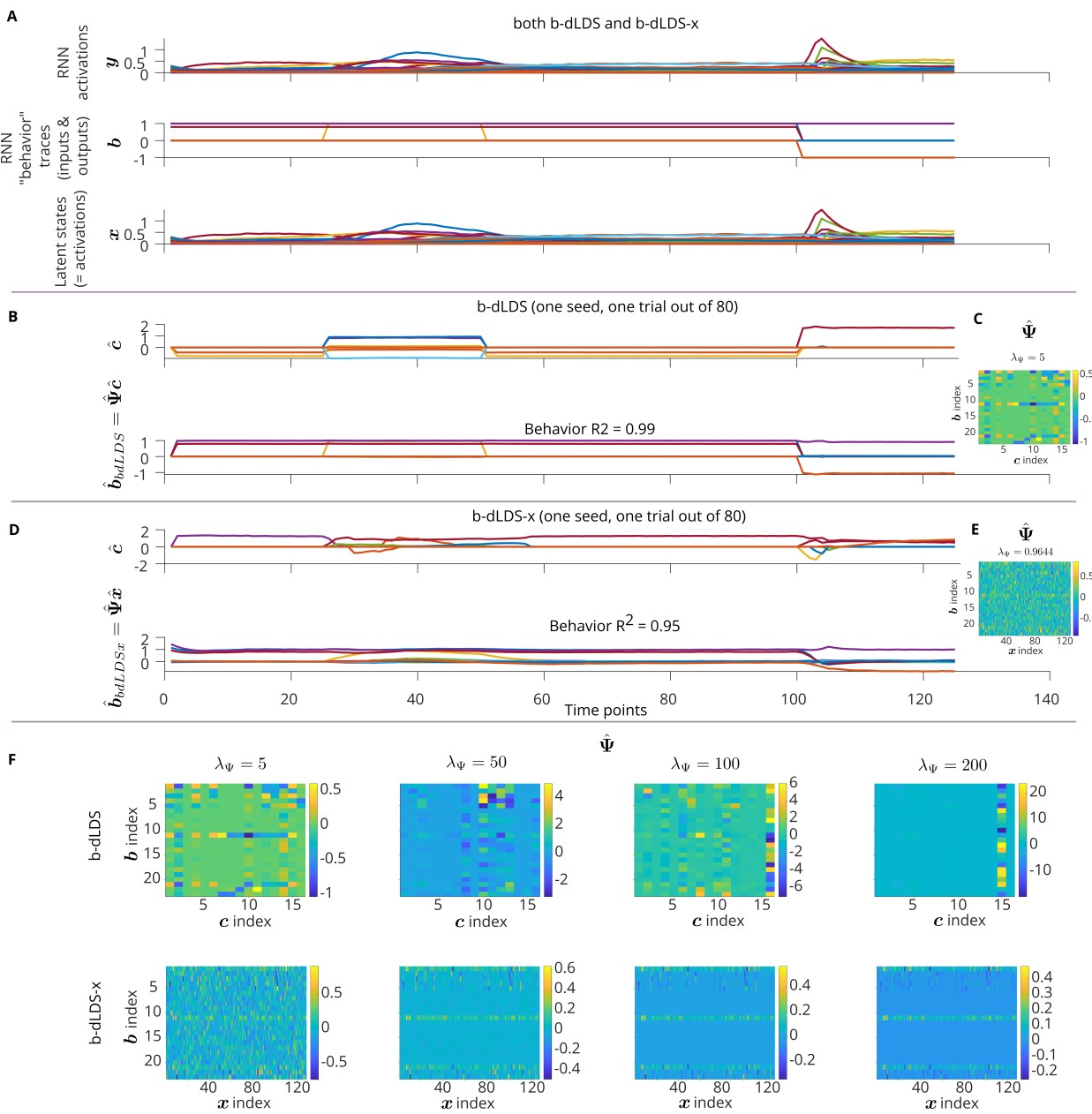

Figure 4: **b-dLDS vs. b-dLDS-x on delayanti task-driven RNN data**. One seed of each model shown. $\boldsymbol{D} = \boldsymbol{I}$. For behavior data reconstruction, b-dLDS $R^2 = 0.99$, while b-dLDS-x $R^2 = 0.95$. The b-dLDS-x behavior reconstruction is constrained to be a linear combination the latent states, while b-dLDS learns a sparser, more interpretable set of dynamics coefficients based on the behavior. **A:** Task-driven RNN model activations, "behavior" traces (23 total inputs and outputs), and latent states (equivalent to the activations because there is no noise in the observations; the activations are known directly). **B:** b-dLDS $\widehat{\boldsymbol{c}}$ and $\widehat{\boldsymbol{b}}$ example. **C:** Corresponding b-dLDS $\widehat{\boldsymbol{\Psi}}$. **D:** b-dLDS-x $\widehat{\boldsymbol{c}}$ and $\widehat{\boldsymbol{b}}$ example. **E:** Corresponding b-dLDS-x $\widehat{\boldsymbol{\Psi}}$. **F:** Comparison of $\widehat{\boldsymbol{\Psi}}$ across regularization strengths on the Frobenius norm for b-dLDS vs. b-dLDS-x.

The model used for the data shown in Figure 5 and Supplementary Figure 6 achieved $R^2 = 0.91$ for neural data reconstruction (over whole time series concatenated; per neuron varies) and $R^2 = 0.97$ for behavior data reconstruction, while the model used for the data shown in Supplementary Figures 7, 8, 9, and 10 achieved $R^2 = 0.93$ for neural data reconstruction and $R^2 = 0.997$ for behavior data reconstruction (over all four behavior time series concatenated).

To characterize the interpretability, reconstruction performance, and consistency of b-dLDS across parameter settings, we ran a series of hyperparameter sensitivity experiments (Supp. Fig. 11,Table 8), ablation comparisons and comparisons against variants of our model (Table 3). For the hyperparameter sensitivity, we found that sufficient model size N was most influential for neural reconstruction, which must be balanced with number of DOs (nF); regularization on the sparsity of dynamics coefficients, when too strong, can cause all of the dynamics coefficients to quickly fall away, also dropping behavior reconstruction $R^2$ to near 0; and there appears to be a knee in $\lambda_{behavior}$ where behavior $R^2$ plateaus. All of these can be tuned with BayesOpt for $R^2$ and median $\widehat{c}$ use.

For the ablation/comparisons we compared b-dLDS to dLDS with *post hoc* behavior correlation (ablating the core modeling addition of b-dLDS), and b-dLDS-x (testing the dynamics coefficients $\boldsymbol{c}$ vs. the latent state $\boldsymbol{x}$ as the prime behaviorally-related model variables). All models had the same latent dimension N=50, which enabled them all to achieve consistent neural $R^2$, and parameters were set using BayesOpt tuning across all models. Using the BayesOpt parameters, b-dLDS-x and b-dLDS were able to achieve comparable behavior reconstruction $R^2$. As with the task-driven RNN example, the difference between the 3 models became stark when considering the correlation between model coefficients and behavior. b-dLDS consistently learned at least one dynamics coefficient that better correlated with behavior $max_i(R^2_{c_i,b})$ than dLDS ($max_i(R^2_{c_i,b})$ or $max_i(R^2_{x_i,b})$). Moreover, b-dLDS's $max_i(R^2_{c_i,b})$ was consistently greater than b-dLDS-x's latent state vs. behavior $max_i(R^2_{x_i,b})$.

Another point of interpretability is the usefulness of $\widehat{\boldsymbol{\Psi}}$. Large values in $\widehat{\boldsymbol{\Psi}}$ indicate dynamics coefficients (or latent states for b-dLDS-x) that strongly contribute to behavior reconstruction We observed that in 7/9 b-dLDS models, the maximum $\widehat{\boldsymbol{\Psi}}$ value mapped to the same index as the most strongly behavior-correlated dynamics coefficient; this only happened in 4/15 b-dLDS-x models. Furthermore, the largest $\widehat{\boldsymbol{\Psi}}$ values were larger in b-dLDS models than in b-dLDS-x models, making these behavior-dynamics relationships more obvious in b-dLDS.

A clear observation from plotting the spatial interactions of the dynamic connectivity maps' strongest connections was that there seemed to be possible asymmetry in the connectivity between the left and right sides of the hindbrain. Prior work has identified asymmetry in the zebrafish fore- and midbrain (Barth et al., 2005; Roussigné et al., 2012; Dreosti et al., 2014; E. et al., 2014; Lekk et al., 2019; Lekk, 2019; Powell et al., 2024; Powell, 2022; Gobbo et al., 2025), especially in regions such as the habenulae, which communicate with the hindbrain, as well as bias in their motor activity (Horstick et al., 2020; Gobbo et al., 2025). While these connections to hindbrain lead to the question of possible asymmetry in the hindbrain, which may be relevant to sensorimotor tasks, asymmetric neural processing in hindbrain has not been reported prior.

To quantify the observed asymmetry and ensure the validity of our observations, we first confirmed several forms of baseline asymmetry in the fluorescence data and in the model outputs. These baselines served to control for any possible impact of asymmetry from the imaging process, etc. (e.g., laser directed from one side of the fish). We calculated the midline of the hindbrain using the weighted centroid of the hindbrain, based on the pixel indices of the neuron locations. (We use the term 'left' to denote pixel indices less than the $y$ axis midline, and 'right' to denote pixel indices greater than the $y$ axis midline.) Then, based on this midline and the centroids of the mapped neural traces, we compared (a) the number of neuron centroids on each side of the midline, (b) the average variance of the delta fluorescence/fluorescence data over time per trace on each side of the midline, and (c) the average variance explained per neuron on each side of the midline (Table 2). Thus, there was indeed asymmetry in both the data, in terms of the spatial distribution of the recorded cells (6% asymmetry), the fluorescence data collected (9% asymmetry; p<0.04 t-test), and the neural reconstruction $R^2$ (17% asymmetry; p<0.001 t-test).

In the b-dLDS zebrafish model (Fig. 5, Supplementary Fig. 6), 6 out of 25 of the DO connectivity maps had an stronger asymmetry in the strongest connections than the baseline asymmetry (from 18% asymmetry in DO 19 to 118% in DO 21). Of these 6 DOs, only one (DO 1) had the third strongest corresponding value in $\boldsymbol{\Psi}$, and only one (DO 19) was among the weakest in $\boldsymbol{\Psi}$. The other dynamics operators that are moderately tied to behavior may have the strongest and most interesting asymmetry, and may be participating in other cognitive processes, as well. Moreover, not all of the DOs were biased toward one side; 5 out of 25 DOs were slightly biased toward the right side. This leads us to conclude that b-dLDS is likely not simply amplifying

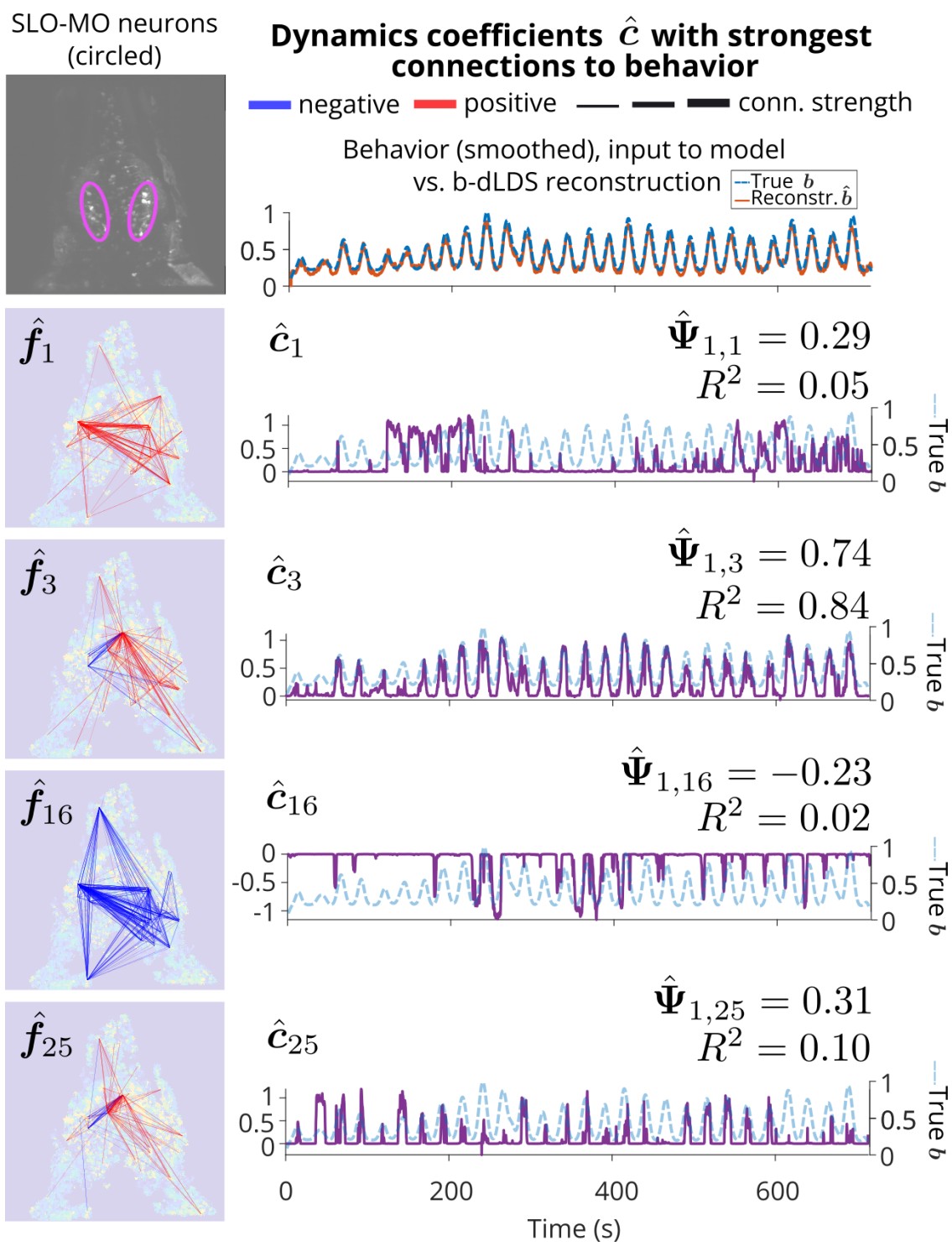

Figure 5: **Four strongest $\Psi$ connections between dynamics coefficients and motor activity in a zebrafish model with one behavior trace.** b-dLDS $R^2 = 0.91$ for neural data reconstruction (over whole time series concatenated; per neuron varies); $R^2 = 0.97$ for behavior data reconstruction. Dynamic connectivity maps for dynamics operators and corresponding dynamics coefficients shown. Motor-aligned dynamics are utilized throughout the recording or a mix of earlier and/or later in the recording.

any baseline asymmetry in due to imaging artifacts. As hindbrain asymmetric processing has not been

previously reported, the identified spatial locations that we identify as focal points for this asymmetry could be a target for future experimentation, especially in mapping the contralateral connections stretching from the anterior hindbrain to near the spinal cord.

We note that the reconstruction $R^2$ baseline, which was the largest baseline by almost a factor of two, might also be impacted by the asymmetry in the functional networks, since the reconstruction $R^2$ is both a function of signal-to-noise and the neural activity itself. Thus the threshold for reporting asymmetry here is relatively conservative, and the actual asymmetry in the hindbrain may be even more pervasive.

Table 2: **b-dLDS connectivity asymmetry in zebrafish hindbrain dynamics.** To form a baseline of asymmetry that might stem from other sources (e.g., imaging artifacts) we compute the asymmetry in the geometry and variance of the fluorescence data itself. These baselines indicate that up to 17% asymmetry might be attributed to other sources. Of the 25 DOs, 6 displayed stronger asymmetry, with more start and/or end points of their connectivity on the left side than on the right.

| Criteria | "Left" side | "Right" side | % asymmetry |
|---|---|---|---|
| # neuron centroids | 2550 | 2400 | 6% |
| Variance of dF/F over time per trace | mean 0.0099, median 0.0052, std 0.0153 | mean 0.0109, median 0.0057, std 0.0174 | 9% (means) |
| $R^2$ per neuron | mean 0.38, median 0.35, std 0.23 | mean 0.46, median 0.43, std 0.27 | 17% (means) |
| # start/end points for largest connections | | | 6 of 25 DOs: 18%-118% |

When we incorporated trial type information into b-dLDS, the model learned different relationships between the dynamics and the motor signal vs. each trial type. While the most strongly motor-related dynamics coefficients were present throughout the recording, the trial type-related dynamics coefficients split up the recordings in time.

The dynamic connectivity maps generated from the DOs corresponding to the dynamics coefficients selected above were corroborated by Yang et al. (2022), who showed that these layers of the hindbrain feature the self-location encoding neurons of the medulla oblongata (SLO-MO) (Fig. 5), which putatively participate in the integration of fish location from optic flow. These dynamic connectivity maps vary in how local vs. hindbrain-wide their connections are. For example, in DO 3 ($R^2_{c_3,b} = 0.84$ and $\mathbf{\Psi}_{1,3} = 0.74$), hindbrain-wide connectivity was highly correlated in time with behavior, with a key node in a posterior region of the hindbrain. Conversely, the connectivity in DOs 1 and 16 showed strong relationships between the SLO-MO regions and more anterior and posterior regions of the hindbrain, with $\widehat{\boldsymbol{c}}_1$ primarily active at the beginning and end of the session, and $\widehat{\boldsymbol{c}}_1$ primarily active in the middle and at the end in the session. Interestingly, the dynamic connectivity maps with the weakest $\mathbf{\Psi}$ coefficients (Supp. Fig. 6) shared some key nodes and connections with the dynamic connectivity maps with the strongest $\mathbf{\Psi}$ coefficients, but the temporal patterns were distinct, highlighting a few sharp, sparse moments of strong recurrent dynamic connectivity both laterally and along the anterior-posterior axis and throughout most of the hindbrain (e.g., DO 11).

## 6 Discussion, Limitations, and Future Work

Here we present behavior-dLDS (b-dLDS), which seeks to identify the complementary behavior-related and behavior-independent subsystems underlying rich whole-brain recordings during behavior. As a dLDS-family model, the learned DOs can be coactive in a variety of linear combinations, describing a highly expressive range of dynamics while maintaining the interpretability of linear systems and the ability to match up specific

dynamics with temporal epochs/behaviors via sparsity. The core novelty of our approach is in treating the behavior as related not to the system *activity* $x_t$ but instead as being related to the expansion coefficients of the dynamics $c_t$, i.e., the time-localized *inter-connectivity*. This distinction innovates on the more typical approach prevalent in modeling natural and artificial systems that are state-centric (Sussillo et al., 2016; Schneider et al., 2023b), whether that be a linear latent state (Sani et al., 2021a; Frandsen et al., 2022) or a nonlinear state defined by, e.g., an autoencoder (Mounayer et al., 2026; Sussillo et al., 2016; Schneider et al., 2023b). Instead we assert that it is not necessarily the latent state that is important to the system's behavior, but how that state is changing over time, i.e., we relate the *trends* in neural activity to *trends* in behavior, rather than a time-point-by-time-point representational model. Moreover, by enabling the partial observability (i.e., that not all systems are behaviorally related), our approach can be broadly relevant to large-scale systems across domains.

While we focus our analysis on the latent dynamics, the latent states $x$ can also help uncover low-dimensional patterns in high-dimensional neural activity. For example, $x$ describes how groups of neurons evolve over time (grouped by $D$). Because $x_t$ is a lower-dimensional representation of the neural activity, it can also be useful for identifying sequential patterns of activation of these groups of neurons over time. Additionally, these groupings can be mapped separately from the dynamic connectivity maps and compared to known circuit components. Thus, $x_t$ could be used by a brain-machine interface for encoding/decoding a sparse version of neural activity. Interestingly, in our testing of b-dLDS-x vs. b-dLDS, we found that the readout of the latent state itself had to be more complex and less interpretable in order to reconstruct the system behavior at the same fidelity. This difference persisted both in synthetic task-driven RNN experiments, as well as the analysis of the zebrafish hindbrain.

In the zebrafish data, b-dLDS was able to tease apart different sub-networks most related to the animal's behavior. These networks revealed significant asymmetry in the functional networks, which, while not reported before in the literature for the hindbrain, matches more generally observed anatomical asymmetry in the zebrafish fore- and mid-brain (Barth et al., 2005; Roussigné et al., 2012; Dreosti et al., 2014; E. et al., 2014; Lekk et al., 2019; Lekk, 2019; Powell et al., 2024; Powell, 2022; Gobbo et al., 2025). The ability of b-dLDS to run on such large datasets ($> 10,000$ neurons) positions it as a potential analysis tool for the growing neural recordings using both imaging (Yang et al., 2022; Marquez-Legorreta et al., 2026) and electrophysiology (Chen et al., 2024a).

A limitation of b-dLDS, like all models that use regularization, is that it can be biased by that regularization. For example, b-dLDS might be biased toward having more or fewer behavior-tied dynamics coefficients by selecting extreme regularization parameter settings for behavior reconstruction or the sparsity of $\Psi$. However, in more moderate regularization parameter setting ranges, the regularization on the behavior vs. dynamics coefficients balance each other, to some extent; the bias toward inferring a sparse set of dynamics coefficients to describe the data counteracts the bias toward perfectly reconstructing the behavior. We explored this via a hyperparameter sensitivity analysis (Appendix Section "Hyperparameter sensitivity experiments - zebrafish and simulation," Supp. Fig. 11, Table 8).

This also brings to mind the question of identifiability vs. consistency. While we can demonstrate a certain level of robustness of our model to changes in initialization (Mudrik et al., 2024a) or hyperparameters, we cannot guarantee that b-dLDS will find "the" unique best decomposition, if one indeed exists for complex, high-dimensional data. This is a challenge with (nearly) any factorization-based method. In order to obtain theoretical guarantees on identifiability, future work will likely draw on the dictionary learning literature (Wu et al., 2018; Hu & Huang, 2023; Cohen & Gillis, 2019).

A key benefit of b-dLDS is the relative computational efficiency: other comparable methods that relate neural dynamics to behavior cannot scale to the tens-of-thousands of neurons needed to model zebrafish data. Both CLDS and PSID encountered memory and size errors, limiting their applicability to the data. This benefit enabled b-dLDS to identify dynamic connectivity maps tied to behavior that revealed the evolution of connections to the SLO-MO region throughout positional homeostasis. As neural datasets continue to grow, models will need to scale even higher, to hundreds of thousands or millions of neurons, in order to map brain-wide circuitry.

b-dLDS complements existing literature on large-scale non-stationary dynamical systems. Generally, contextual dynamical models are a broad class, including, e.g., models of community interactions during an infectious disease outbreak (Garfinkel; Lessler et al., 2016), the impact of pharmacological interventions (Brunton et al., 2016; Viknesh et al., 2026), user activity on a social media platform (Jouyban et al., 2025), vehicles in a traffic network (Meneguzzer, 2022), climate trends (Christensen & Berner, 2019), or RNN activations Driscoll et al. (2024); Xie et al. (2024), as in Figure 4. While these models sometimes include explicit state transitions, dynamical systems models such as b-dLDS could enable the modeling of nonstationary processes, continuous state combinations, and dynamics tied to external variables. For example, in pharmacokinetics, b-dLDS can complement methods such as SINDy (Brunton et al., 2016; Viknesh et al., 2026), a nonlinear dynamical systems identification method, which features explicit mathematical priors that b-dLDS does not. b-dLDS's ability to deal with high-dimensional data could be especially useful as more body systems are simultaneously modeled.

In conclusion, b-dLDS proposes a model in which the relationship between brain activity and behavior is driven not by the latent state representing the instantaneous neural activity, but a representation of the dynamical system itself (i.e., which interactions are driving the system). While the architecture is based on an existing model, dLDS, our addition of behavior to the model (1) serves a unique role in the space of current models of neural activity and behavior, prioritizing *partial* behavior regularization of *dynamics*, and (2) provides a new way to relate the dynamics that dLDS-family models learn to behavior (in comparison to the current process of *post hoc* analysis). To emphasize this point, our comparisons to variants of dLDS, including *post hoc* analysis of "vanilla" dLDS coefficients and dLDS where the behavior is modeled as a function of the latent state $x$ (b-dLDS-x) show, on both synthetic and real datasets) that b-dLDS maintains improved interpretability.

b-dLDS can model 14,000-dimensional zebrafish hindbrain calcium imaging data and identify which latent neural dynamics are most likely behavior-generating. This approach enables future work analyzing dynamic connectivity at key time points to compare internal computations to active swimming. The modularity of the b-dLDS framework could also be used to create new dLDS-family models with hierarchical relationships between observed data, e.g., sensory inputs, and latent model components in the style of b-dLDS.

## Acknowledgments

EYezerets and ASC were supported in part by NSF CAREER award 2340338. MBA and EYang were supported by the Howard Hughes Medical Institute. We would like to acknowledge Sue Ann Koay for the original version of the manifold illustration in Figure 1, Victor Geadah for his help setting up CLDS, Maryam Shanechi for her insights into PSID, Bryan Tseng's help setting up PSID, and Jessica Ye's help getting started with the task-driven RNN data from Driscoll et al. (2024).

## Data Availability

Zebrafish data is available upon request from En Yang (enyang@unc.edu) or Misha Ahrens (ahrensm@janelia.hhmi.org). Zebrafish results and simulation data are available at `https://osf.io/syvx5/`. RNN data is available from Driscoll et al. (2024).

## Code Availability

The code is available at `https://github.com/NeuralCoDy/bdLDS/`.

## Competing Interests

The authors declare no competing interests.

## Impact Statement

This paper introduces a machine learning method intended for decoding neuroscience and behavior data, and may impact other fields with time series data, as well. We do not anticipate any specific societal consequences of note.

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

## A Dynamic Connectivity Mapping

Dynamic connectivity (Yezerets et al., 2025) is calculated by mapping individual DOs or linear combinations of DOs into the ambient space:

$$Connectivity = \boldsymbol{D}\boldsymbol{F}\boldsymbol{D}^T. \tag{14}$$

Dynamic connectivity is a description of the correlation between individual units (e.g., neurons) states $\boldsymbol{y}$ over time, i.e., the outer product of $\boldsymbol{y}_t$ and $\boldsymbol{y}_{t-1}$, $\mathbb{E}[\boldsymbol{y}_t \boldsymbol{y}_{t-1}^T]$. Under the b-dLDS model, we can compute this correlation through the latent state correlations between $\boldsymbol{x}_t$ and $\boldsymbol{x}_{t-1}$ mapped back to the ambient space. Given $\boldsymbol{y}_t = \boldsymbol{D}\boldsymbol{x}_t$ and $\boldsymbol{x}_t = \boldsymbol{F}\boldsymbol{x}_{t-1}$,

$$
\begin{aligned}
\mathbb{E}[(\boldsymbol{y}_t - \boldsymbol{\mu}_y)(\boldsymbol{y}_{t-1} - \boldsymbol{\mu}_y)^T] &= \mathbb{E}[(\boldsymbol{D}(\boldsymbol{x}_t - \boldsymbol{\mu}_x))(\boldsymbol{D}(\boldsymbol{x}_{t-1} - \boldsymbol{\mu}_x))^T] & (15)\\
&= \mathbb{E}[\boldsymbol{D}\boldsymbol{x}_t \boldsymbol{x}_{t-1}^T \boldsymbol{D}^T] - \mathbb{E}[\boldsymbol{D}\boldsymbol{\mu}_x]\mathbb{E}[\boldsymbol{D}\boldsymbol{\mu}_x]^T & (16)\\
&= \mathbb{E}[\boldsymbol{D}]\mathbb{E}[(\boldsymbol{F}\boldsymbol{x}_{t-1} + \boldsymbol{\epsilon})\boldsymbol{x}_{t-1}^T]\mathbb{E}[\boldsymbol{D}^T] - \boldsymbol{D}\boldsymbol{\mu}_x\boldsymbol{\mu}_x^T\boldsymbol{D}^T & (17)\\
&= \mathbb{E}[\boldsymbol{D}](\mathbb{E}[\boldsymbol{F}\boldsymbol{x}_{t-1}\boldsymbol{x}_{t-1}^T] + \mathbb{E}[\boldsymbol{\epsilon}_x]\mathbb{E}[\boldsymbol{x}_{t-1}^T])\mathbb{E}[\boldsymbol{D}^T] - \boldsymbol{D}\boldsymbol{\mu}_x\boldsymbol{\mu}_x^T\boldsymbol{D}^T & (18)\\
&= \mathbb{E}[\boldsymbol{D}]\mathbb{E}[\boldsymbol{F}\boldsymbol{x}_{t-1}\boldsymbol{x}_{t-1}^T]\mathbb{E}[\boldsymbol{D}^T] - \boldsymbol{D}\boldsymbol{\mu}_x\boldsymbol{\mu}_x^T\boldsymbol{D}^T & (19)\\
&= \boldsymbol{D}\mathbb{E}[\boldsymbol{F}]\mathbb{E}[\boldsymbol{x}_{t-1}\boldsymbol{x}_{t-1}^T]\boldsymbol{D}^T - \boldsymbol{D}\boldsymbol{\mu}_x\boldsymbol{\mu}_x^T\boldsymbol{D}^T & (20)\\
&= \boldsymbol{D}\boldsymbol{F}\boldsymbol{\Sigma}_x\boldsymbol{D}^T - \boldsymbol{D}\boldsymbol{\mu}_x\boldsymbol{\mu}_x^T\boldsymbol{D}^T & (21)\\
&= \boldsymbol{D}\left(\boldsymbol{F}\boldsymbol{\Sigma}_x - \boldsymbol{\mu}_x\boldsymbol{\mu}_x^T\right)\boldsymbol{D}^T, & (22)
\end{aligned}
$$

where $\boldsymbol{\mu}_x = \mathbb{E}[\boldsymbol{x}]$ is the expected value of the latent variables, $\mathbb{E}[\boldsymbol{\epsilon}_x] = \boldsymbol{0}$, and $\boldsymbol{F}_t = \sum_{m=1}^M \boldsymbol{f}_m c_{mt}$. Note that we can write Equation 20 from Equation 19 since $\boldsymbol{c}_t$ are *i.i.d.* random Laplacian vectors and thus each $\boldsymbol{c}_t$ is independent of $\boldsymbol{x}_{t-1}$, implying that $\mathbb{E}[\boldsymbol{F}\boldsymbol{x}_{t-1}\boldsymbol{x}_{t-1}^T] = \mathbb{E}[\boldsymbol{F}]\mathbb{E}[\boldsymbol{x}_{t-1}\boldsymbol{x}_{t-1}^T]$.

Under the assumption that (1) the latent states are isotropically distributed over time, we can further approximate $\Sigma_x \approx I$, and (2) the latent states are centered at zero, i.e., $\boldsymbol{\mu}_x =, 0$, we can simplify the above expression to

$$\mathbb{E}[\boldsymbol{y}_t \boldsymbol{y}_{t-1}^T] \approx \boldsymbol{D}\boldsymbol{F}\boldsymbol{D}^T. \tag{23}$$

In cases where the above assumptions are violated, the appropriate corrections in Equation 22 can be applied.

Here we define this connectivity such that each entry $i, j$ in each $\boldsymbol{f}$ included in $\boldsymbol{F}$ represents the influence of group $j$ in the latent space at time $t-1$ on group $i$ in the latent space at time $t$. Thus, connections go from columns to rows. As a result, when plotting, we transpose this connectivity matrix.

## B Experiment Details

### B.1 Simulation of two independent systems and behavior

As in Figure 5 in Mudrik et al. (2024a), we test behavior-dLDS on a simulation in which two independent groups of time traces are generated from two non-overlapping sets of operators, unit-normed and applied repetitively over random durations. No more than two operators are active at the same time, and no more than one operator is active for each independent group at the same time. We vary the Gaussian noise levels applied after generating the dynamics operators and dynamics coefficients. (In the "added noise" simulation in Supplementary Figure 3, we test one arbitrary noise level, 0.1. We add noise with mean 0 and variance 1 to the generated latent states data, scaled by the square root of the noise level setting, in order to achieve a noise variance of 0.1.) Note that $\boldsymbol{D} = \boldsymbol{I}$. In this behavior-dLDS simulation, we also generate simulated behavior $\boldsymbol{b}_t$ from the dynamics coefficients using a randomly generated $\boldsymbol{\Psi}$, where $\boldsymbol{b}_t = \boldsymbol{\Psi}\boldsymbol{c}_t$. (In order to make $\boldsymbol{\Psi}$ only map a subset of the dynamics coefficients, other columns were set to 0 after random initialization.) Supplementary Figures 1, 2, and 3 are shown here; Figure 2 is shown above. The CLDS comparison continues in Supplementary Figure 4.

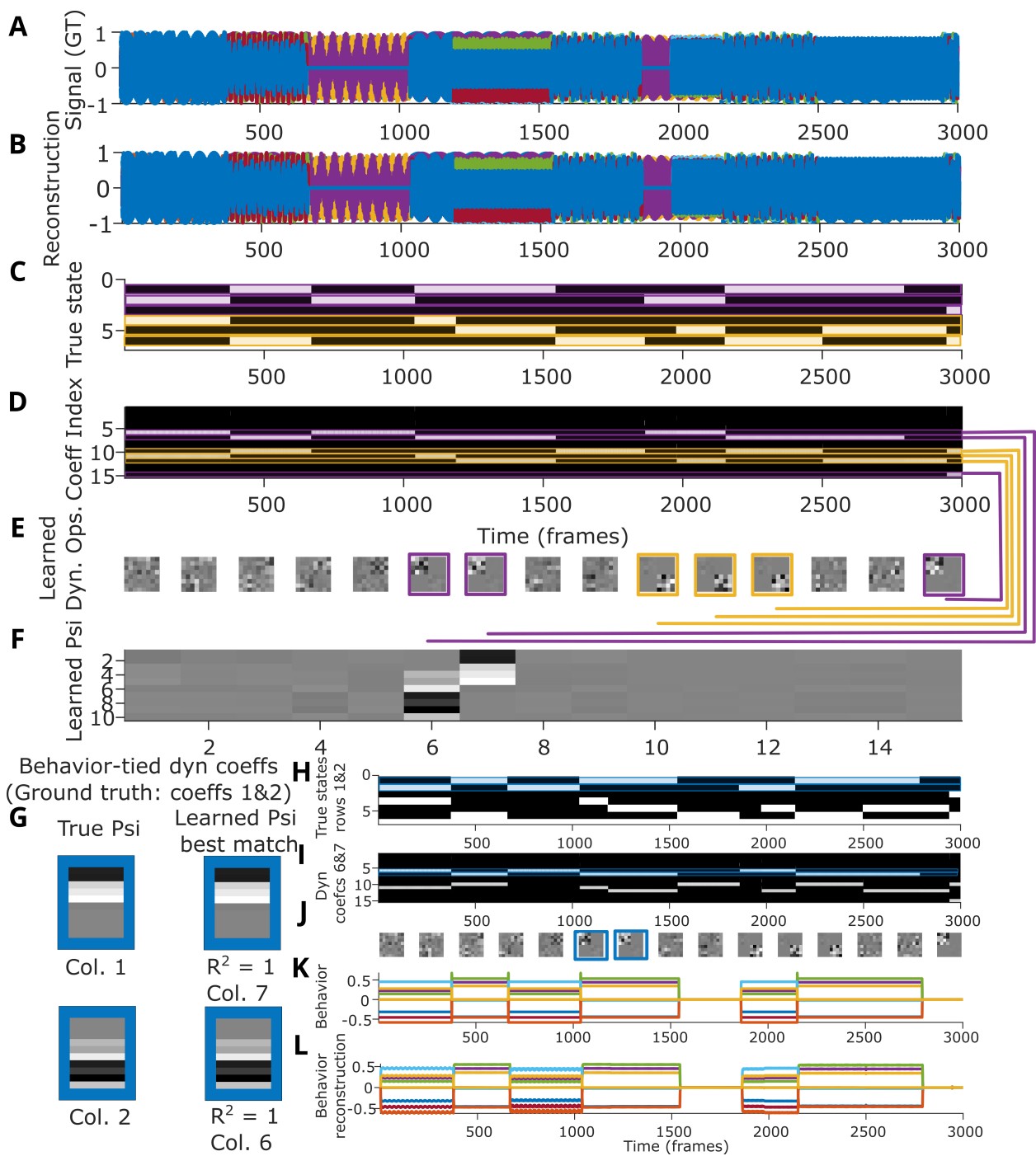

Supplementary Figure 1: **Two-independent-systems simulation with 10 behaviors generated from the first two ground truth states, with some overlap in their combined effects. A:** Ground truth signal fed into the model. **B:** Signal reconstruction from b-dLDS. **C:** True dynamics coefficients used to generate the data. **D:** Learned dynamics coefficients (absolute value). **E:** Learned dynamics operators (1-15). **F:** Learned $\Psi$. **G:** Best reconstruction of the true $\Psi$ (2 columns, other columns all 0). **H:** Highlighted dynamics coefficient used to simulate behavior data. **I:** Highlighted dynamics coefficient corresponding to best match of learned $\Psi$. **J:** Highlighted corresponding dynamics operator. **K:** Simulated behavior ground truth. **L:** Behavior reconstruction.

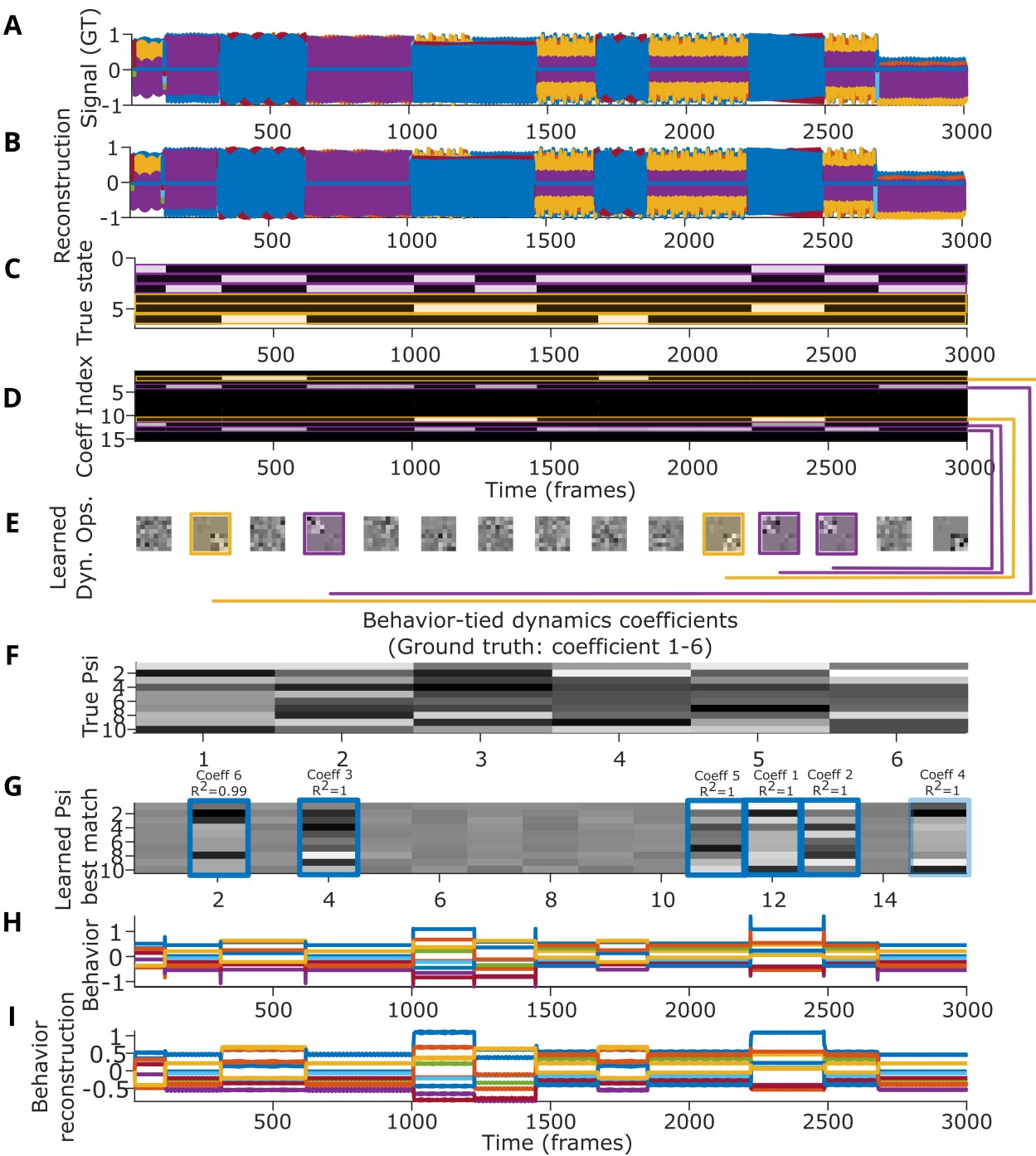

Supplementary Figure 2: **Two-independent-systems simulation with 10 behaviors generated from a combination of all 6 ground truth states.** **A:** Ground truth signal fed into the model. **B:** Signal reconstruction from b-dLDS. **C:** True dynamics coefficients used to generate the data - one was randomly initialized as all zeros. **D:** Learned dynamics coefficients (absolute value). **E:** Learned dynamics operators (1-15). **F:** True $\Psi$. **G:** Best reconstruction of the true $\Psi$ (6 columns). **H:** True behavior. **I:** Behavior reconstruction.

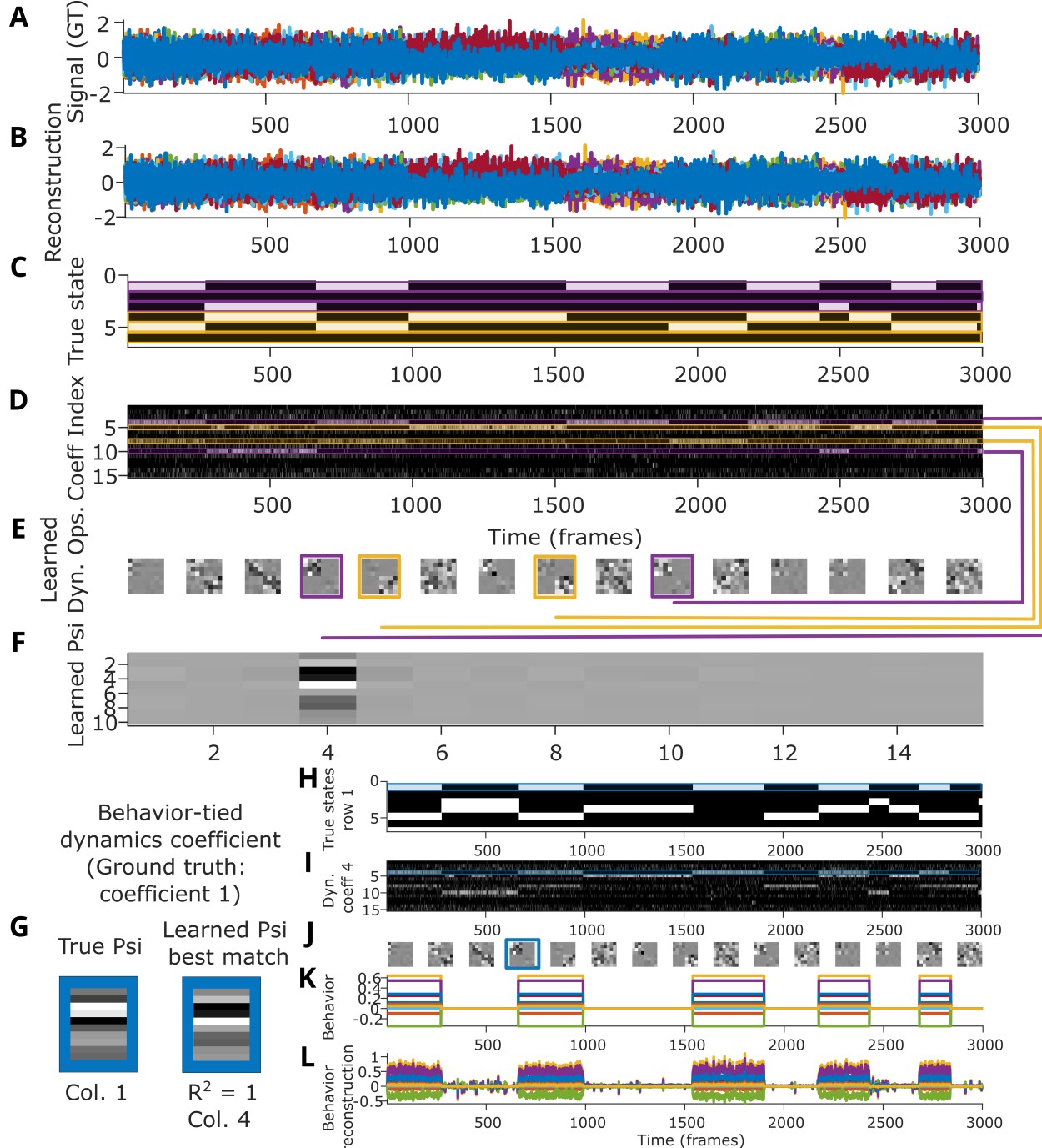

Supplementary Figure 3: **Two-independent-systems simulation with 10 behaviors generated from the first ground truth state, with added Gaussian noise in the "neural" data. A:** Ground truth signal fed into the model, with Gaussian noise scaled by $\sqrt{0.1}$. **B:** Signal reconstruction from b-dLDS. **C:** True dynamics coefficients used to generate the data - two were randomly initialized as all zeros. **D:** Learned dynamics coefficients (absolute value). **E:** Learned dynamics operators (1-15). **F:** Learned $\boldsymbol{\Psi}$. **G:** Best reconstruction of the true $\boldsymbol{\Psi}$ (1 column, other columns all 0). **H:** Highlighted dynamics coefficient used to simulate behavior data. **I:** Highlighted dynamics coefficient corresponding to best match of learned $\boldsymbol{\Psi}$. **J:** Highlighted corresponding dynamics operator. **K:** Simulated behavior ground truth. **L:** Behavior reconstruction.

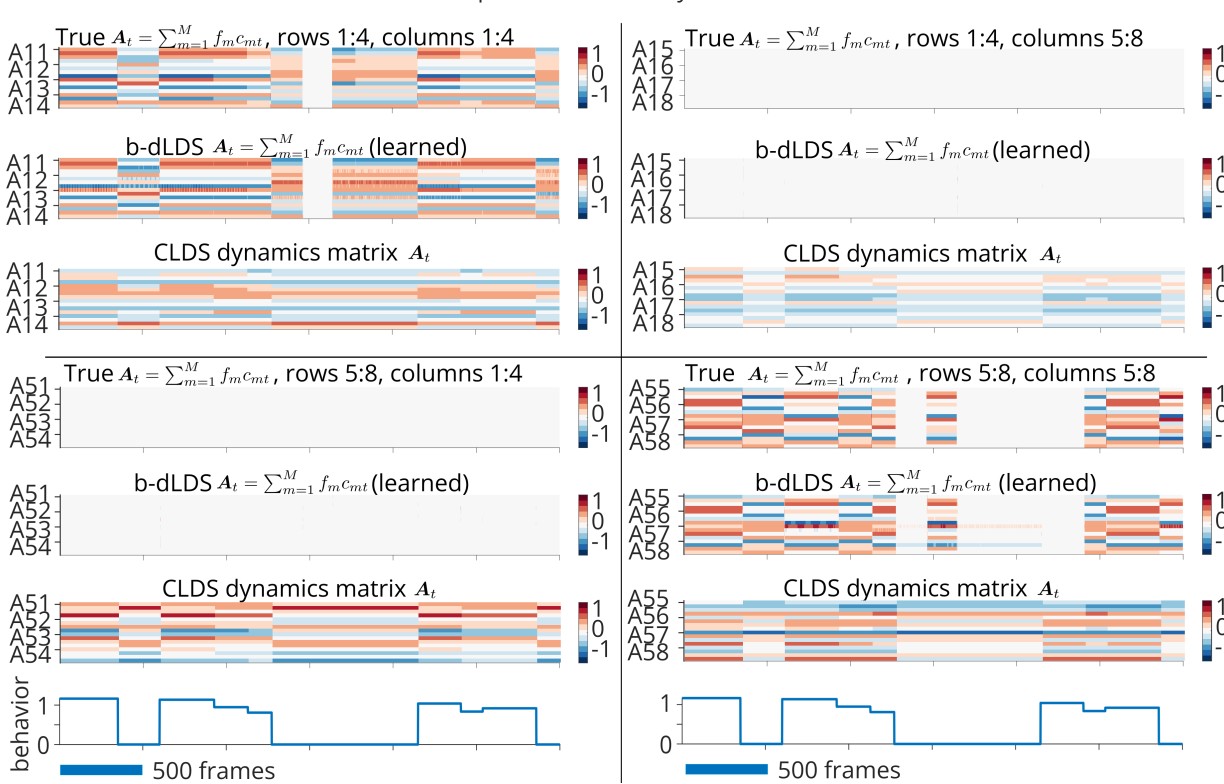

Supplementary Figure 4: **CLDS vs. b-dLDS learned dynamics, continued.** CLDS learns an 8x8 dynamics matrix $\boldsymbol{A}_t$, while b-dLDS learns $\boldsymbol{f}$ and $\boldsymbol{c}$, which can be combined via $\sum_{m=1}^{M} f_m c_{mt}$ to create an equivalent matrix. We compare each block of the matrix (top left 4x4 through bottom right 4x4) to the ground truth dynamics from the simulation. b-dLDS closely matches the ground truth and correctly learns that the two off-diagonal corners are 0, indicating 2 independent systems; CLDS instead incorrectly follows the behavior in all dynamics. Each row of each heatmap is an entry of the 4x4 block of the overall 8x8 matrix (A12 means row 1 column 2, 16 rows per heatmap).

## B.2  Zebrafish data

This calcium imaging data was recorded across 9 slices of a single larval zebrafish (*Danio rerio* hindbrain at 6 Hz using lightsheet microscopy by En Yang (Yang et al., 2022) in order to find the self-location encoding medulla oblongata neurons (SLO-MO) (14090 neural time traces, 5123 time points; 13098 neural time traces and 4282 time points used after preprocessing in experiment with 1 motor signal (calculated based on electrical recordings from a cluster of neurons as described by Yang et al. (2022)); 14064 neural time traces and 4200 time points used after preprocessing in experiment with 1 motor signal and 3 trial type labels). SLO-MO is hypothesized to form a circuit with the inferior olive, cerebellum, and optic tectum to enable the fish to maintain its position in a current (positional homeostasis).

### B.2.1  Zebrafish data preprocessing

Based on feedback from the Ahrens lab, extreme fluorescence values were capped at +/- 1 a.u. Neural activity was median filtered with window size 5 and behavior was smoothed with a Gaussian with $\sigma = 20$, based on the approximate swim trial durations. The data was then rescaled to calculate delta fluorescence/fluorescence, centered on the mode of each trace and rescaled by a robust standard deviation metric for each trace. A soft normalization of 1 was added to the dF/F rescaling, which proved crucial for improving reconstruction for this and other large-scale zebrafish lightsheet data with b-dLDS. The Ljung-Box test was used to filter out traces with insignificant autocorrelation over a large lag (500 to 1000 time points), since signal traces should show autocorrelation throughout the session given the repetitive nature of the positional homeostasis task, while noise traces should have rapidly decaying autocorrelation. Only the last 4282 out of 5123 time points (after the first 5 complete swim trials) were used for modeling the data with 1 behavior (just the motor output as behavior fed into the model), and 4200 time points for the models with 4 "behaviors" (motor plus trial types, trimming the last trial).

### B.2.2  Trial types

Trial types are determined by the virtual reality displacement of the fish during the hold (open loop) period between each fictive swimming response. This hold period occurred within the first 54 time points of each 150-time-point-long trial, after which the fish was exposed to a backward current in closed loop and could perform fictive swimming against it. There were three displacement patterns, which appear as a series of positive (forward) and negative (backward) displacements in the visual velocity data that accompanies this dataset. Trial type 1 refers to 1 negative (backward) displacement, followed by a delay, followed by 3 negative displacements in rapid succession. Trial type 2 refers to 1 forward displacement, followed by a delay, followed by 3 negative displacements. Trial type 3 refers to a delay, followed by 3 negative displacements.

While trial types are an input rather than a behavior output from the fish, we include them as "behavior" information to the model because they are relevant to the behavior structure.

### B.3  What if b-dLDS mapped behavior to the latent states?

In Table 3 we present a model comparison of b-dLDS on zebrafish data to dLDS, which could be considered an ablation of $\lambda_{behavior}$, and b-dLDS-x, which maps behavior to the latent states $\boldsymbol{x}$ instead of the dynamics coefficients $\boldsymbol{c}$. The default versions of these models do not use step size rescaling and the Frobenius norm on $\boldsymbol{\Psi}$, but models are also shown with these settings turned on ("reg on Psi"). The b-dLDS-x settings are set using the same settings as b-dLDS in the first version of b-dLDS-x in this table, but we also compare this to b-dLDS-x with $\lambda_0$ and $\lambda_4$ (i.e., regularization on $\left\| \boldsymbol{x}_t - \widetilde{\boldsymbol{F}}_t \boldsymbol{c}_t \right\|_2^2$ and $\left\| \boldsymbol{b}_t - \boldsymbol{\Psi} \boldsymbol{x}_t \right\|_2^2$, in the case of b-dLDS-x), optimized for maximum behavior reconstruction $R^2$ via Bayesian Optimization (BayesOpt - MATLAB). BayesOpt was run for 30 setting combinations with 100 iterations of b-dLDS-x each time to find the best combination of selected hyperparameters (it is possible that more optimization could be done). All models shown in the table were run for 5000 iterations. Model settings are found in Table 4.

We found that b-dLDS provides a better and more concise behavior fit in the latent space than dLDS or b-dLDS-x. For b-dLDS models, the strongest $\widehat{\boldsymbol{\Psi}}$ value had the same index as the dynamics coefficient with

the greatest $R^2$ with behavior in 7/9 cases. In contrast, for b-dLDS-x, the strongest $\widehat{\Psi}$ value had the same index as the latent state $\widehat{x}$ with the greatest $R^2$ with behavior in only 4/15 cases. Moreover, the values of the max $\widehat{c}$ $R^2$ with behavior were 0.69-0.95 for b-dLDS, whereas the values of the max $\widehat{x}$ $R^2$ with behavior were much lower, at 0.26-0.42. (This was even when nF and N were equal - see nF=50 models.)

Table 3: **b-dLDS, dLDS, and b-dLDS-x model comparison, with runtimes and memory utilization.** b-dLDS-x represents an alternative to b-dLDS where the behavior maps to the latent states rather than to the dynamics coefficients. With additional optimization, b-dLDS-x can match b-dLDS in overall behavior reconstruction. However, b-dLDS provides better interpretability: its max $\Psi$ values are larger, and so are its max $R^2$ values between the dynamics coefficients and behavior, compared to the max $R^2$ values between the dynamics coefficients and behavior for b-dLDS-x. In the two versions of b-dLDS shown here, two times out of three, these two indices (max $\Psi$ and max $c$ vs. $b$ $R^2$ also corresponded; in contrast, they only corresponded in one version of b-dLDS-x,"b-dLDS-x with reg on $\Psi$", which also had worse behavior reconstruction (see "Max $\Psi$ value" and "Idx max $\Psi$" vs. "Max $R^2$, $c$ vs. $b$" and "Idx max $c$" for b-dDLS models, and "Max $\Psi$ value" and "Idx max $\Psi$" vs. "Max $R^2$, $x$ vs. $b$" and "Idx max $x$" for b-dLDS-x models). Please note that runtime and memory utilization are only meant to be a rough estimate, as these were dependent on other processes happening in parallel on the server, and there is no built-in function for calculating memory utilization for MATLAB on Linux machines; instead we rely on subtracting start from end memory used, sometimes resulting in negative values (Dvorak, 2026).

| Model | Seed | $R_y^2$ | $R_b^2$ | Max $\widehat{\Psi}$ value | Idx max $\widehat{\Psi}$ | Max $R^2$, $\widehat{c}$ vs. $b$ | Idx max $\widehat{c}$ | Max $R^2$ $\widehat{x}$ vs. $b$ | Idx max $\widehat{x}$ | Time (s) | RAM (KB) |
|---|---|---|---|---|---|---|---|---|---|---|---|
| b-dLDS | 1 | 0.91 | 0.96 | 0.68 | 25 | 0.82 | 25 | 0.36 | 3 | 6.11E+04 | -2.17E+08 |
| | 2 | 0.9 | 0.95 | 0.29 | 20 | 0.84 | 25 | 0.36 | 43 | 6.09E+04 | -2.18E+08 |
| | 3 | 0.91 | 0.97 | 0.74 | 3 | 0.84 | 3 | 0.33 | 21 | 4.74E+04 | 7.91E+07 |
| dLDS | 1 | 0.91 | | | | 0.14 | 22 | 0.33 | 49 | 6.08E+04 | -2.17E+08 |
| | 2 | 0.91 | | | | 0.19 | 15 | 0.3 | 50 | 6.09E+04 | -2.18E+08 |
| | 3 | 0.91 | | | | 0.11 | 1 | 0.28 | 36 | 4.71E+04 | 9.34E+07 |
| b-dLDS-x | 1 | 0.91 | 0.89 | 0.45 | 21 | 0.19 | 7 | 0.32 | 49 | 4.71E+04 | 4.64E+07 |
| | 2 | 0.91 | 0.91 | 0.29 | 1 | 0.16 | 8 | 0.31 | 5 | 2.74E+04 | 5.72E+07 |
| | 3 | 0.91 | 0.92 | 0.27 | 26 | 0.15 | 9 | 0.35 | 36 | 2.74E+04 | 5.77E+07 |
| b-dLDS with reg on $\Psi$ | 1 | 0.9 | 0.96 | 0.76 | 8 | 0.69 | 8 | 0.26 | 43 | 2.38E+04 | 3.57E+07 |
| | 2 | 0.9 | 0.96 | 0.79 | 9 | 0.8 | 9 | 0.39 | 18 | 2.38E+04 | 3.60E+07 |
| | 3 | 0.91 | 0.96 | 0.37 | 25 | 0.73 | 15 | 0.28 | 42 | 2.64E+04 | 6.80E+07 |
| b-dLDS-x with reg on $\Psi$ | 1 | 0.91 | 0.89 | 0.3 | 36 | 0.16 | 24 | 0.35 | 36 | 2.40E+04 | 3.56E+07 |
| | 2 | 0.91 | 0.89 | 0.33 | 20 | 0.31 | 23 | 0.32 | 20 | 2.65E+04 | 5.79E+07 |
| | 3 | 0.9 | 0.89 | 0.28 | 9 | 0.15 | 14 | 0.35 | 9 | 2.65E+04 | 5.82E+07 |
| b-dLDS-x, optim. for $R_b^2$ with BayesOpt | 1 | 0.91 | 0.97 | 0.55 | 16 | 0.26 | 20 | 0.42 | 12 | 3.66E+04 | 2.67E+07 |
| | 2 | 0.91 | 0.96 | 0.43 | 7 | 0.14 | 13 | 0.41 | 31 | 2.84E+04 | 5.43E+07 |
| | 3 | 0.91 | 0.97 | 0.41 | 34 | 0.16 | 21 | 0.37 | 11 | 3.66E+04 | 1.76E+07 |
| b-dLDS-x, optim. for $R_b^2$ with BayesOpt | 1 | 0.91 | 0.93 | 0.37 | 45 | 0.16 | 1 | 0.38 | 50 | 3.39E+04 | -4.56E+07 |
| | 2 | 0.9 | 0.94 | 0.43 | 16 | 0.17 | 23 | 0.4 | 13 | 2.90E+04 | 5.48E+07 |
| | 3 | 0.9 | 0.94 | 0.21 | 32 | 0.14 | 13 | 0.34 | 25 | 2.90E+04 | 5.51E+07 |

and reg on $\boldsymbol{\Psi}$

| Model | seed | | | | | | | | | | |
|---|---|---|---|---|---|---|---|---|---|---|---|
| dLDS, nF = 50 | 1 | 0.90 | | | | 0.13 | 30 | 0.36 | 1 | 7.42E+04 | -3.85E+07 |
| | 2 | 0.91 | | | | 0.11 | 23 | 0.34 | 43 | 7.44E+04 | -3.59E+07 |
| | 3 | 0.91 | | | | 0.16 | 43 | 0.29 | 26 | 7.40E+04 | -3.63E+07 |
| b-dLDS, nF = 50, with reg on $\boldsymbol{\Psi}$ | 1 | 0.91 | 0.99 | 0.94 | 50 | 0.95 | 50 | 0.38 | 22 | 7.43E+04 | -3.91E+07 |
| | 2 | 0.91 | 0.98 | 0.94 | 42 | 0.92 | 42 | 0.29 | 34 | 7.42E+04 | -3.30E+07 |
| | 3 | 0.91 | 0.99 | 0.94 | 31 | 0.94 | 31 | 0.30 | 31 | 7.41E+04 | -3.57E+07 |
| b-dLDS-x, nF = 50, BayesOpt (previous settings), with reg on $\boldsymbol{\Psi}$ | 1 | 0.91 | 0.94 | 0.34 | 44 | 0.13 | 18 | 0.45 | 44 | 7.48E+04 | -3.44E+07 |
| | 2 | 0.91 | 0.94 | 0.24 | 33 | 0.13 | 16 | 0.34 | 27 | 7.44E+04 | -3.78E+07 |
| | 3 | 0.92 | 0.94 | 0.40 | 47 | 0.10 | 19 | 0.37 | 50 | 7.49E+04 | -3.67E+07 |

| Summary avg. (3 seeds) | | | | # match (idx max $\widehat{\boldsymbol{\Psi}}$ vs. idx max $\widehat{\boldsymbol{c}}$ or $\widehat{\boldsymbol{x}}$ $R^2$) | | | | |
|---|---|---|---|---|---|---|---|---|
| b-dLDS | 0.91 | 0.96 | 0.57 | 2 | 0.83 | 0.35 | 5.65E+04 | -1.19E+08 |
| dLDS | 0.91 | | | | 0.15 | 0.30 | 5.63E+04 | -1.14E+08 |
| b-dLDS-x | 0.91 | 0.91 | 0.34 | 0 | 0.17 | 0.33 | 3.40E+04 | 5.37E+07 |
| b-dLDS, $\boldsymbol{\Psi}$ reg | 0.90 | 0.96 | 0.64 | 2 | 0.74 | 0.31 | 2.47E+04 | 4.66E+07 |
| b-dLDS-x, $\boldsymbol{\Psi}$ reg | 0.91 | 0.89 | 0.30 | 3 | 0.21 | 0.34 | 2.57E+04 | 5.06E+07 |
| b-dLDS-x, BO | 0.91 | 0.97 | 0.46 | 0 | 0.19 | 0.4 | 3.38E+04 | 3.29E+07 |
| b-dLDS-x, BO, $\boldsymbol{\Psi}$ reg | 0.90 | 0.94 | 0.34 | 0 | 0.16 | 0.37 | 3.06E+04 | 2.14E+07 |
| dLDS, nF 50 | 0.91 | | | | 0.13 | 0.33 | 7.42E+04 | -3.69E+07 |
| b-dLDS, nF 50, $\boldsymbol{\Psi}$ reg | 0.91 | 0.99 | 0.94 | 3 | 0.94 | 0.32 | 7.42E+04 | -3.59E+07 |
| b-dLDS-x, nF 50, BO, $\boldsymbol{\Psi}$ reg | 0.91 | 0.94 | 0.33 | 1 | 0.12 | 0.39 | 7.47E+04 | -3.63E+07 |

Table 4: **Meaning of key b-dLDS parameters, heuristics for setting them, and settings used for model comparisons in Table 3:** one motor signal input as behavior to b-dLDS and b-dLDS-x.

| Parameter | Meaning | How to set | Settings used for zebrafish models (in order of appearance in Table 3) |
|---|---|---|---|
| nF | number of dynamics operators (DOs) | Based on models used in zebrafish figures. Also tried matching nF to N for fairness of $\widehat{\boldsymbol{x}}$ and $\widehat{\boldsymbol{c}}$ vs. $\boldsymbol{b}$ $R^2$ comparisons. | 25 (first 7 model types); 50 (last 3 model types, "nF=50") |

| N | number of latent dimensions | Based on models used in zebrafish figures. | 50 |
|---|---|---|---|
| lambda_val | regularization: sparsity of states ($\widehat{\boldsymbol{x}}$) | Based on models used in zebrafish figures. | 0.05 |
| lambda_history | regularization: smoothness of states ($\widehat{\boldsymbol{x}}$), but more specifically, reconstruction of states from dynamics operators and dynamics coefficients | Based on models used in zebrafish figures (with exceptions for b-dLDS-x BayesOpt). | 0.23; 0.1154 (b-dLDS-x BayesOpt model types) |
| lambda_b | regularization: sparsity of dynamics coefficients ($\widehat{\boldsymbol{c}}$) | Based on models used in zebrafish figures. | 0.05 |
| lambda_historyb | regularization: smoothness of dynamics coefficients ($\widehat{\boldsymbol{c}}$) | Based on models used in zebrafish figures. | 0 |
| max_iters | max number of iterations | Based on models used in zebrafish figures. | 5000 |
| F_update | T/F: update dynamics operators | Based on models used in zebrafish figures. | 1 |
| D_update | T/F: update observation matrix $\widehat{\boldsymbol{D}}$ | Based on models used in zebrafish figures. | 1 |
| N_ex | number of samples | Based on models used in zebrafish figures. | 40 |
| T_s | number of time points per sample | As above | 30 |
| lambda_f | Based on models used in zebrafish figures. | 0.05 | |
| solver_type | 'fista' or 'tfocs' | Based on models used in zebrafish figures. | 'tfocs' |
| behaviordLDS | Use behavior data to regularize b-dLDS model? | 1 or 0 | 1 for b-dLDS and b-dLDS-x, 0 for dLDS |
| verysparsebhv | $\widehat{\boldsymbol{\Psi}}$ update step size rescaling | Baseline zebrafish model did not use this. Only used for model types "reg on $\boldsymbol{\Psi}$". Not relevant for dLDS. | 1 (model type "reg on $\boldsymbol{\Psi}$") (default: 0) |

| | | | |
|---|---|---|---|
| step_psi | Gradient descent step size for updating $\widehat{\boldsymbol{\Psi}}$ | Based on models used in zebrafish figures. Not relevant for dLDS. | 10 |
| lambda_behavior | regularization: reconstruction of behavior from dynamics coefficients and $\widehat{\boldsymbol{\Psi}}$ | Based on models used in zebrafish figures (except b-dLDS-x with BayesOpt). Not relevant for dLDS. | 1; 1.9953 (b-dLDS-x BayesOpt) |
| psinorm | norm for $\widehat{\boldsymbol{\Psi}}$ update step | Baseline zebrafish model did not use this. Only used for model types "reg on $\boldsymbol{\Psi}$". In that case, used Frobenius norm. Not relevant for dLDS. | "frob" (model type "reg on $\boldsymbol{\Psi}$") (default: "norm") |
| lambda2 | regularization: sparsity on $\widehat{\boldsymbol{\Psi}}$ (Frobenius only) | Based on models used in zebrafish figures (except for last 2 models: BayesOpt). Only used for model types labeled as "reg on $\boldsymbol{\Psi}$". Not relevant for dLDS. | 5 |

## B.4 behavior-dLDS parameters and model selection

### B.4.1 Simulations

Most key model parameters (Table 5) were maintained from the dLDS simulation in Figure 5 of the dLDS paper (Mudrik et al., 2024a). However, model fitting was drastically improved by implementing stronger regularization on the orthogonality of the dynamics operators ($\lambda_f$) that decays more quickly. While we initially used BayesOpt to find the optimal regularization on the behavior reconstruction ($\lambda_4 = \lambda_{behavior}$), we found that using the same small $\lambda$ as we used for the zebrafish models was a better balance of modeling the dynamics and learning the correct $\boldsymbol{\Psi}$. Increasing the number of samples and time points per sample also improved reconstruction. Large step size of $\boldsymbol{\Psi}$ helped to explore the space of $\boldsymbol{\Psi}$ faster. Parameters were tested iteratively to achieve $R^2_{\boldsymbol{\Psi}}$ near 1. Supplementary Figure 5 demonstrates the impact of including the Frobenius norm and step size rescaling on the $\widehat{\boldsymbol{\Psi}}$ update.

Table 5: **Meaning of key b-dLDS parameters, heuristics for setting them, and settings used for 4 simulation model types:** one dynamics coefficient maps to all 10 behaviors; 2 dynamics coefficients map to all 10 behaviors; all dynamics coefficients map to all 10 behaviors; and one dynamics coefficient maps to all 10 behaviors, with added Gaussian noise in the simulated data.

| Parameter | Meaning | How to set | Settings used for 4 simulation types |
|---|---|---|---|

| nF | number of dynamics operators (DOs) | First set to 2x the number of PCs required for 95% variance explained. Then decrease until the median number of active dynamics coefficients starts decreasing. (Goal: minimize the number of DOs while maintaining the inherent dimensionality of the data.) Alternative heuristic: use the number of known behavioral states as a benchmark and double it to get the starting point for the number of DOs, and then start decreasing from there. | 15 |
|---|---|---|---|
| N | number of latent dimensions | Known based on data, given that $D = I$. | 8 |
| lambda_val | regularization: sparsity of states $\widehat{x}$ | Increase to reduce the number of states active at the same time (moderately sensitive) | 1.0000e-04 |
| lambda_history | regularization: smoothness of states $\widehat{x}$, but more specifically, reconstruction of states from dynamics operators and dynamics coefficients | Increase to encourage states to change smoothly from time point to time point (less sensitive) | 1.0000e-04 |
| lambda_b | regularization: sparsity of dynamics coefficients (c) | Increase to reduce the number of DOs active at the same time (sensitive) | 0.25 |
| lambda_historyb | regularization: smoothness of dynamics coefficients (c) | Increase to encourage DO coefficients to change smoothly from time point to time point | 0.45 |
| max_iters | max number of iterations | Increase if dF not converging to near 0, decrease if converging quickly to save time | 500;5000;500;500;5000 (Fig. 3) |
| F_update | T/F: update dynamics operators | True because operators must be learned here | 1 |
| D_update | T/F: update observation matrix $D$ | False because $D = I$ here | 0 |
| N_ex | number of samples | More samples (and time points) can better learn the data but increases the time required per iteration | 100;100;200;200;100 (Fig. 3) |

| T_s | number of time points per sample | As above | 200 |
|---|---|---|---|
| step_d | gradient step size of $\widehat{\boldsymbol{D}}$ update | Not relevant, not used because $\boldsymbol{D} = \boldsymbol{I}$ here, so D_update is off | 1 - doesn't matter |
| step_f | gradient step size of $\boldsymbol{f}$ dictionary update | Start small | 10 |
| step_decay | gradient step decay | Start near 1 | 0.99995 |
| lambda_f | regularization: promotes diversity of dynamics operators | sensitive, start near 0 | 0.1 |
| lambda_f_decay | reduces lambda_f gradually as dynamics operators are learned | Start near 1 | 0.996 |
| solver_type | 'fista' or 'tfocs' | As the name suggests, fista is significantly faster | 'fista' |
| special | '' or 'no_obs' | Choose 'no_obs' when $\boldsymbol{D} = \boldsymbol{I}$ | 'no_obs' |
| behaviordLDS | Use behavior data to regularize b-dLDS model? | 1 or 0 | 1 |
| step_psi | Gradient descent step size for updating $\boldsymbol{\Psi}$ | larger step size explores the space in fewer iterations | 100;10 (Fig. 3) |
| lambda_behavior | regularization: reconstruction of behavior from dynamics coefficients and $\boldsymbol{\Psi}$ | start small | 0.1 |
| verysparsebhv | option for improved dictionary update of $\boldsymbol{\Psi}$ where some dynamics coefficients may not map to behavior at all, resulting in zero values and NaN errors ("sparse suspected") | 1 or 0 (default) | 1 |

| | | | |
|---|---|---|---|
| psinorm | option for improved update of $\boldsymbol{\Psi}$ - Frobenius norm pushes some dynamics coefficients mappings to 0 | 'norm' (default) or 'frob' | 'frob' |
| lambda2 | scaling: Frobenius norm term in dictionary update | start small | 5 |

### B.4.2 Task-driven RNN

The delayanti task is described in Driscoll et al. (2024). 64 out of 80 test trials were used to learn the model parameters, and coefficients were inferred on all 80 test trials after the model was learned, with 3 random seeds (random 64 trials, random model initialization). One trial from one seed for each model type is shown in Figure 4. Settings were optimized via BayesOpt - MATLAB to maximize behavior reconstruction, with 100 iterations of b-dLDS or 10 iterations of b-dLDS-x (oddly, 100 iterations produced an equivalent behavior $R^2$ but zeroed out all dynamics coefficients) and 30 BayesOpt iterations used for parameter optimization. 500 iterations were used for final models shown in figures. Settings for b-dLDS and b-dLDS-x can be found in Table 6.

Table 6: **Meaning of key b-dLDS and b-dLDS-x parameters, heuristics for setting them, and settings used for the task-driven RNN experiments.**

| Parameter | Meaning | How to set | Settings used (if differ, first b-dLDS setting shown; then b-dLDS-x) |
|---|---|---|---|
| nF | number of dynamics operators (DOs) | Set to 2x the number of PCs required for 95% variance explained. | 16 |
| N | number of latent dimensions | Known based on data, given that $\boldsymbol{D} = \boldsymbol{I}$. | 128 |
| lambda_val | regularization: sparsity of states ($\widehat{\boldsymbol{x}}$) | Optimize with BayesOpt. | 0.9772;0.0293 |
| lambda_history | regularization: smoothness of states ($\widehat{\boldsymbol{x}}$), but more specifically, reconstruction of states from dynamics operators and dynamics coefficients | Optimize with BayesOpt. | 0.8702;0.0636 |

| lambda_b | regularization: sparsity of dynamics coefficients ($\widehat{c}$) | Optimize with BayesOpt. | 0.25908;0.0315 |
|---|---|---|---|
| lambda_historyb | regularization: smoothness of dynamics coefficients ($\widehat{c}$) | Keep small; not tuned for these experiments. | 1.00e-05 |
| max_iters | max number of iterations | Increase if dF not converging to near 0, decrease if converging quickly to save time | 500 |
| F_update | T/F: update dynamics operators | True because operators must be learned here | 1 |
| D_update | T/F: update observation matrix $\widehat{D}$ | False because $D = I$ here | 0 |
| N_ex | number of samples | Not tuned; short trial duration, 1 sample, full trial length. | 1 |
| T_s | number of time points per sample | As above | 125 |
| step_d | gradient step size of $\widehat{D}$ update | Not relevant, not used because $D = I$ here, so D_update is off | 1 - doesn't matter |
| step_f | gradient step size of $f$ dictionary update | Start small; not tuned for these experiments | 10 |
| step_decay | gradient step decay | Start near 1; not tuned for these experiments | 0.99995 |
| lambda_f | regularization: promotes diversity of dynamics operators | sensitive, start near 0; not tuned for these experiments | 1.00e-05 |
| lambda_f_decay | reduces lambda_f gradually as dynamics operators are learned | Start near 1; not tuned for these experiments | 0.996 |
| solver_type | 'fista' or 'tfocs' | As the name suggests, fista is significantly faster | 'fista' |
| special | '' or 'no_obs' | Choose 'no_obs' when $D = I$ | 'no_obs' |
| behaviordLDS | Use behavior data to regularize b-dLDS model? | 1 or 0 | 1 |
| step_psi | Gradient descent step size for updating $\widehat{\Psi}$ | Optimize with BayesOpt; larger step size explores the space in fewer iterations | 10; 17 |

| | | | |
|---|---|---|---|
| lambda_behavior | regularization: reconstruction of behavior from dynamics coefficients and $\widehat{\boldsymbol{\Psi}}$ | Optimize with BayesOpt; start small; not relevant to b-dLDS-x in the $\boldsymbol{D} = \boldsymbol{I}$ case (NoObs does not include $\boldsymbol{x}$ in the LASSO step, so $\lambda_{behavior}$ is not used | 0.9313 (b-dLDS only) |
| verysparsebhv | option for improved dictionary update of $\boldsymbol{\Psi}$ where some dynamics coefficients may not map to behavior at all, resulting in zero values and NaN errors ("sparse suspected") | 1 or 0 (default) - refer to ablation experiments | 1 |
| psinorm | option for improved update of $\widehat{\boldsymbol{\Psi}}$ - Frobenius norm pushes some dynamics coefficients mappings to 0 | 'norm' (default) or 'frob' - refer to ablation experiments | 'frob' |
| lambda2 | scaling: Frobenius norm term in dictionary update | Optimize with BayesOpt; start small | 5; 0.9644 |

### B.4.3 Zebrafish

dLDS and behavior-dLDS reconstruction $R^2$ proved particularly sensitive to model size. This was best approximated by the number of principal components required by PCA to achieve an $R^2$ around 0.8-0.9. Thus, we opted to use 50 latent dimensions. We then started with 50 dynamics operators and scaled down from there to 25 operators without loss of $R^2$ or median number of active operators at a time. Regularization coefficients were optimized starting from the coefficients used to model *C. elegans* data in Yezerets et al. (2025). The additional sparsity settings for updating $\boldsymbol{\Psi}$, e.g., the Frobenius norm, were not used. Settings can be found in Table 7.

Table 7: **Meaning of key b-dLDS parameters, heuristics for setting them, and settings used for two zebrafish model types in Figure 5 and Supplementary Figures 6, 7, 8, 9, and 10:** one motor signal input as behavior, or one motor signal plus three trial types as one-hot encodings.

| Parameter | Meaning | How to set | Settings used for zebrafish models (1 vs. 4 behavior inputs) |
|---|---|---|---|
| | | | |

| nF | number of dynamics operators (DOs) | First set to 2x the number of PCs required for 95% variance explained. Then decrease until the median number of active dynamics coefficients starts decreasing. (Goal: minimize the number of DOs while maintaining the inherent dimensionality of the data.) Alternative heuristic: use the number of known behavioral states as a benchmark and double it to get the starting point for the number of DOs, and then start decreasing from there. | 25 |
|---|---|---|---|
| N | number of latent dimensions | Set to at least the number of PCs required for 80-90% variance explained. | 50 |
| lambda_val | regularization: sparsity of states $\widehat{x}$ | Increase to reduce the number of states active at the same time (moderately sensitive) | 0.05 |
| lambda_history | regularization: smoothness of states $(\widehat{x})$, but more specifically, reconstruction of states from dynamics operators and dynamics coefficients | Increase to encourage states to change smoothly from time point to time point (less sensitive) | 0.23 |
| lambda_b | regularization: sparsity of dynamics coefficients $\widehat{c}$ | Increase to reduce the number of DOs active at the same time (sensitive) | 0.05 |
| lambda_historyb | regularization: smoothness of dynamics coefficients $\widehat{c}$ | Increase to encourage DO coefficients to change smoothly from time point to time point | 0 |
| max_iters | max number of iterations | Increase if dF not converging to near 0, decrease if converging quickly to save time | 5000 |
| F_update | T/F: update dynamics operators | True because operators must be learned here | 1 |
| D_update | T/F: update observation matrix $\widehat{D}$ | True because $D \neq I$ here | 1 |
| N_ex | number of samples | More samples (and time points) can better learn the data but increases the time required per iteration | 40 |

| T_s | number of time points per sample | As above | 30 |
|---|---|---|---|
| lambda_f | regularization: promotes diversity of dynamics operators | sensitive, start near 0 | 0.05 |
| solver_type | 'fista' or 'tfocs' | Although fista is faster to run, tfocs was used for neural data previously in (Yezerets et al., 2025) and produces more regularly-timed dynamics coefficients on neural data | 'tfocs' |
| behaviordLDS | Use behavior data to regularize b-dLDS model? | 1 or 0 | 1 |
| step_psi | Gradient descent step size for updating $\widehat{\boldsymbol{\Psi}}$ | larger step size explores the space in fewer iterations | 10 |
| lambda_behavior | regularization: reconstruction of behavior from dynamics coefficients and $\widehat{\boldsymbol{\Psi}}$ | start small | 1 |

### B.4.4 Hyperparameter optimization

Due to the number of parameters, the full space was not explored via grid search, but rather determined heuristically for each parameter as they were added onto the base model, partially based on previous results on neural data (Yezerets et al., 2025) and simulations (Mudrik et al., 2024a). In terms of initializing the right number of dynamics operators, Mudrik et al. (2024a) showed that dLDS-family models can be initialized with more than enough DOs, and those that are not needed can be suppressed through regularization. Hyperparameters were initially explored via BayesOpt (BayesOpt - MATLAB). It was observed on a larger neural dataset (60,000 neurons, not shown here) that $\boldsymbol{y}$ reconstruction $R^2$ was particularly constrained by the number of latent dimensions $\boldsymbol{x}$, depending on the true latent dimensionality of the data: 118 resulted in $R^2 = 0.6$, 398 dimensions improved $R^2$ to 0.7, and 500 dimensions improved $R^2$ to about 0.75. The PCA $R^2$ curve can be used to give model size selection (i.e., select the number of latent dimensions needed based on the "maximum achievable" $R^2$ that that many PCA dimensions would provide). See also Appendix Section "Zebrafish data preprocessing". In future work, we hope to establish a fast, algorithmic way to narrow down appropriate parameter settings for a particular dataset.

### B.5 Hyperparameter sensitivity experiments - zebrafish and simulation

We have swept the following parameters, which must be tuned to achieve acceptable neural and behavior $R^2$ and reasonable dynamics coefficient utilization throughout the trial, as well as model size and number of dynamics coefficients mapped to the behavior: N (number of latent dimensions), nF (number of DOs), the regularization parameter for sparsity of $\widehat{\boldsymbol{c}}$, and the regularization parameter for $\boldsymbol{b}$ reconstruction (Supp. Fig. 11, Table 8). In the zebrafish models, increasing N improved neural $R^2$ until a plateau was reached; increasing nF increased median $\widehat{\boldsymbol{c}}$ use, with a small plateau at nF = 20 to 25. Surprisingly, neural $R^2$ *decreased* with increased nF. As far as regularization on the sparsity of $\widehat{\boldsymbol{c}}$, neural and behavior $R^2$ and median $\widehat{\boldsymbol{c}}$ use remained robust to small perturbations, but regularization that was too strong (e.g., 1.05) suppressed all of

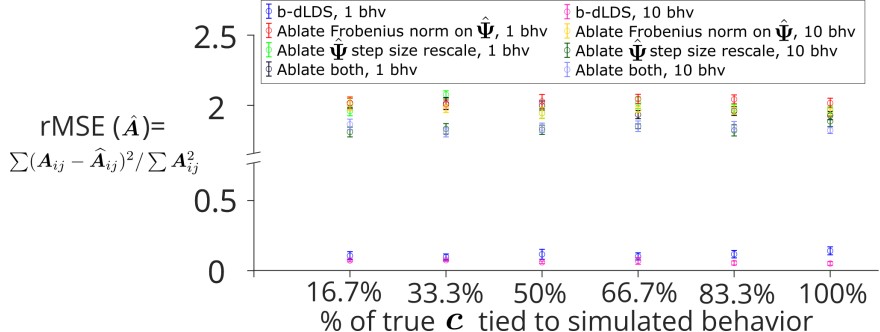

Supplementary Figure 5: **Ablation of regularization on $\widehat{\boldsymbol{\Psi}}$ update step.** b-dLDS with these settings optimized (same simulation setup as shown in Figure 3K) achieves lower relative mean squared error on dynamics reconstruction than ablated models, regardless of the number of behavior traces (1 or 10) or number of dynamics tied to the behavior traces (behavior traces are generated as a linear combination of the simulated dynamics traces).

the dynamics coefficients, which also dramatically affected behavior $R^2$. Median $\widehat{\boldsymbol{c}}$ use increased moderately (between median $\widehat{\boldsymbol{c}}$ use = 5 or 6) alongside $\lambda_{behavior}$; unsurprisingly, behavior $R^2$ strongly followed $\lambda_{behavior}$, with some appearance of a plateau after approximately $\lambda_{behavior} = 1$. Thus, N impacts neural $R^2$, and nF, $\lambda_{dyn.coef}$, and $\lambda_{behavior}$ must balance each other on the number of active dynamics coefficients, with particular sensitivity to overregularization via $\lambda_{dyn.coef}$. The simulations tell a similar story: too strong of a $\lambda_{dyn.coef}$ can reduce model performance, while stronger $\lambda_{behavior}$ tends to improve model performance. Moreover, in these simulations, we know the median $\widehat{\boldsymbol{c}}$ use should be between 1 and 2, since up to two DOs can be active at a time, and usually are. Regularization parameter settings close to those used in Figures 3 and 5 earlier in the paper demonstrated model outputs that are robust to perturbations within a small range.

Table 8: **Hyperparameter sensitivity experiments.** * indicates settings used in paper figures. Values to test were chosen based on values used in figures, plus 0.01, 0.1, and 1, at least. The results shown are not necessarily converged to their maximum potential $R^2$, for example, because only 500 iterations were run for each of these tests (rather than the 5000 used in zebrafish figures).

| Model | Param. | Val. | Median $\widehat{\boldsymbol{c}}$ use | Mean | $R_b^2$ | Mean | $R_y^2$ | Mean | * Setting used |
|---|---|---|---|---|---|---|---|---|---|
| Zebrafish | N | 25 | 7 | 6.67 | 0.98 | 0.97 | 0.73 | 0.72 | |
| hindbrain | | 25 | 6 | | 0.98 | | 0.70 | | |
| | | 25 | 7 | | 0.95 | | 0.72 | | |
| | | 35 | 6 | 6.00 | 0.97 | 0.97 | 0.78 | 0.78 | |
| | | 35 | 6 | | 0.99 | | 0.78 | | |
| | | 35 | 6 | | 0.97 | | 0.79 | | |
| | | 45 | 6 | 6.00 | 0.97 | 0.98 | 0.83 | 0.84 | |
| | | 45 | 6 | | 0.99 | | 0.84 | | |
| | | 45 | 6 | | 0.98 | | 0.84 | | |
| | | 50 | 6 | 6.00 | 0.97 | 0.98 | 0.87 | 0.86 | * |
| | | 50 | 6 | | 0.98 | | 0.86 | | |
| | | 50 | 6 | | 0.98 | | 0.86 | | |
| | | 55 | 6 | 6.33 | 0.97 | 0.98 | 0.88 | 0.87 | |

| | | | | | | | | | |
|---|---|---|---|---|---|---|---|---|---|
| | | 55 | 7 | | 0.98 | | 0.88 | | |
| | | 55 | 6 | | 0.98 | | 0.86 | | |
| Zebrafish hindbrain | nF | 5 | 2 | 3.00 | 0.96 | 0.97 | 0.88 | 0.87 | |
| | | 5 | 4 | | 0.98 | | 0.86 | | |
| | | 5 | 3 | | 0.98 | | 0.86 | | |
| | | 10 | 5 | 4.33 | 0.98 | 0.97 | 0.86 | 0.86 | |
| | | 10 | 4 | | 0.96 | | 0.86 | | |
| | | 10 | 4 | | 0.97 | | 0.86 | | |
| | | 20 | 5 | 5.33 | 0.98 | 0.98 | 0.86 | 0.86 | |
| | | 20 | 6 | | 0.98 | | 0.85 | | |
| | | 20 | 5 | | 0.98 | | 0.87 | | |
| | | 25 | 6 | 5.33 | 0.98 | 0.98 | 0.86 | 0.86 | * |
| | | 25 | 5 | | 0.98 | | 0.86 | | |
| | | 25 | 5 | | 0.98 | | 0.86 | | |
| | | 50 | 7 | 7.00 | 0.98 | 0.98 | 0.85 | 0.85 | |
| | | 50 | 7 | | 0.98 | | 0.86 | | |
| | | 50 | 7 | | 0.98 | | 0.84 | | |
| Zebrafish hindbrain | $\lambda_{behavior}$ | 0.5 | 5 | 5.33 | 0.90 | 0.90 | 0.87 | 0.87 | |
| | | 0.5 | 6 | | 0.94 | | 0.87 | | |
| | | 0.5 | 5 | | 0.85 | | 0.87 | | |
| | | 1 | 6 | 5.67 | 0.99 | 0.98 | 0.86 | 0.86 | * |
| | | 1 | 5 | | 0.98 | | 0.85 | | |
| | | 1 | 6 | | 0.98 | | 0.86 | | |
| | | 1.01 | 6 | 6.00 | 0.98 | 0.97 | 0.88 | 0.87 | |
| | | 1.01 | 6 | | 0.95 | | 0.86 | | |
| | | 1.01 | 6 | | 0.97 | | 0.86 | | |
| | | 1.1 | 5 | 5.67 | 0.99 | 0.98 | 0.85 | 0.85 | |
| | | 1.1 | 6 | | 0.97 | | 0.86 | | |
| | | 1.1 | 6 | | 0.98 | | 0.85 | | |
| | | 2 | 6 | 6.00 | 0.99 | 0.99 | 0.87 | 0.87 | |
| | | 2 | 6 | | 0.99 | | 0.86 | | |
| | | 2 | 6 | | 1.00 | | 0.87 | | |
| Zebrafish hindbrain | $\lambda_{dyn.coef.}$ | 0.05 | 6 | 5.67 | 0.98 | 0.98 | 0.85 | 0.86 | * |
| | | 0.05 | 5 | | 0.98 | | 0.87 | | |
| | | 0.05 | 6 | | 0.97 | | 0.86 | | |
| | | 0.051 | 6 | 6.33 | 0.97 | 0.97 | 0.87 | 0.86 | |
| | | 0.051 | 7 | | 0.98 | | 0.87 | | |
| | | 0.051 | 6 | | 0.96 | | 0.86 | | |
| | | 0.06 | 5 | 5.00 | 0.98 | 0.98 | 0.87 | 0.86 | |
| | | 0.06 | 5 | | 0.98 | | 0.86 | | |
| | | 0.06 | 5 | | 0.98 | | 0.86 | | |
| | | 0.15 | 2 | 1.33 | 0.97 | 0.98 | 0.87 | 0.86 | |
| | | 0.15 | 1 | | 0.98 | | 0.86 | | |
| | | 0.15 | 1 | | 0.98 | | 0.86 | | |
| | | 1.05 | 0 | 0.00 | 0.06 | 0.06 | 0.88 | 0.87 | |
| | | 1.05 | 0 | 0.67 | 0.06 | 0.24 | 0.87 | | |
| | | 1.05 | 0 | 1.33 | 0.06 | 0.54 | 0.87 | | |
| Simulation (as in Figure 3) | $\lambda_{behavior}$ | 0.1 | 2 | 2.00 | 0.59 | 0.79 | | | * |
| | | 0.1 | 2 | | 0.98 | | | | |
| | | 0.1 | 2 | | NaN | | | | |
| | | 0.101 | 2 | 2.00 | 0.98 | 0.80 | | | |
| | | 0.101 | 2 | | 0.44 | | | | |
| | | 0.101 | 2 | | 0.99 | | | | |

| | | | | | | | |
|---|---|---|---|---|---|---|---|
| | | 0.11 | 2 | 2.00 | 0.97 | 0.98 | |
| | | 0.11 | 2 | | 0.99 | | |
| | | 0.11 | 2 | | 0.99 | | |
| | | 0.2 | 2 | 1.67 | 0.99 | 0.99 | |
| | | 0.2 | 2 | | 0.99 | | |
| | | 0.2 | 1 | | 0.99 | | |
| | | 1.1 | 3 | 2.67 | 1.00 | 0.99 | |
| | | 1.1 | 3 | | NaN | 0.98 | |
| | | 1.1 | 2 | | 0.99 | 0.98 | |
| Simulation | $\lambda_{dyn.coef.}$ | 0.25 | 1 | 1.67 | 0.96 | 0.98 | * |
| (as in | | 0.25 | 2 | | 0.97 | | |
| Figure 3) | | 0.25 | 2 | | 0.99 | | |
| | | 0.251 | 2 | 2.00 | 0.95 | 0.96 | |
| | | 0.251 | 2 | | 0.97 | | |
| | | 0.251 | 2 | | 0.97 | | |
| | | 0.26 | 2 | 1.67 | 0.98 | 0.98 | |
| | | 0.26 | 2 | | 0.98 | | |
| | | 0.26 | 1 | | 0.99 | | |
| | | 0.35 | 2 | 2.00 | 0.98 | 0.96 | |
| | | 0.35 | 2 | | 0.97 | | |
| | | 0.35 | 2 | | 0.93 | | |
| | | 1.35 | 1 | 0.67 | 0.79 | 0.70 | |
| | | 1.35 | 1 | | 0.64 | | |
| | | 1.35 | 0 | | 0.67 | | |

## B.6 Computing infrastructure

All models shown were run on an internal cluster. Only CPUs were used.

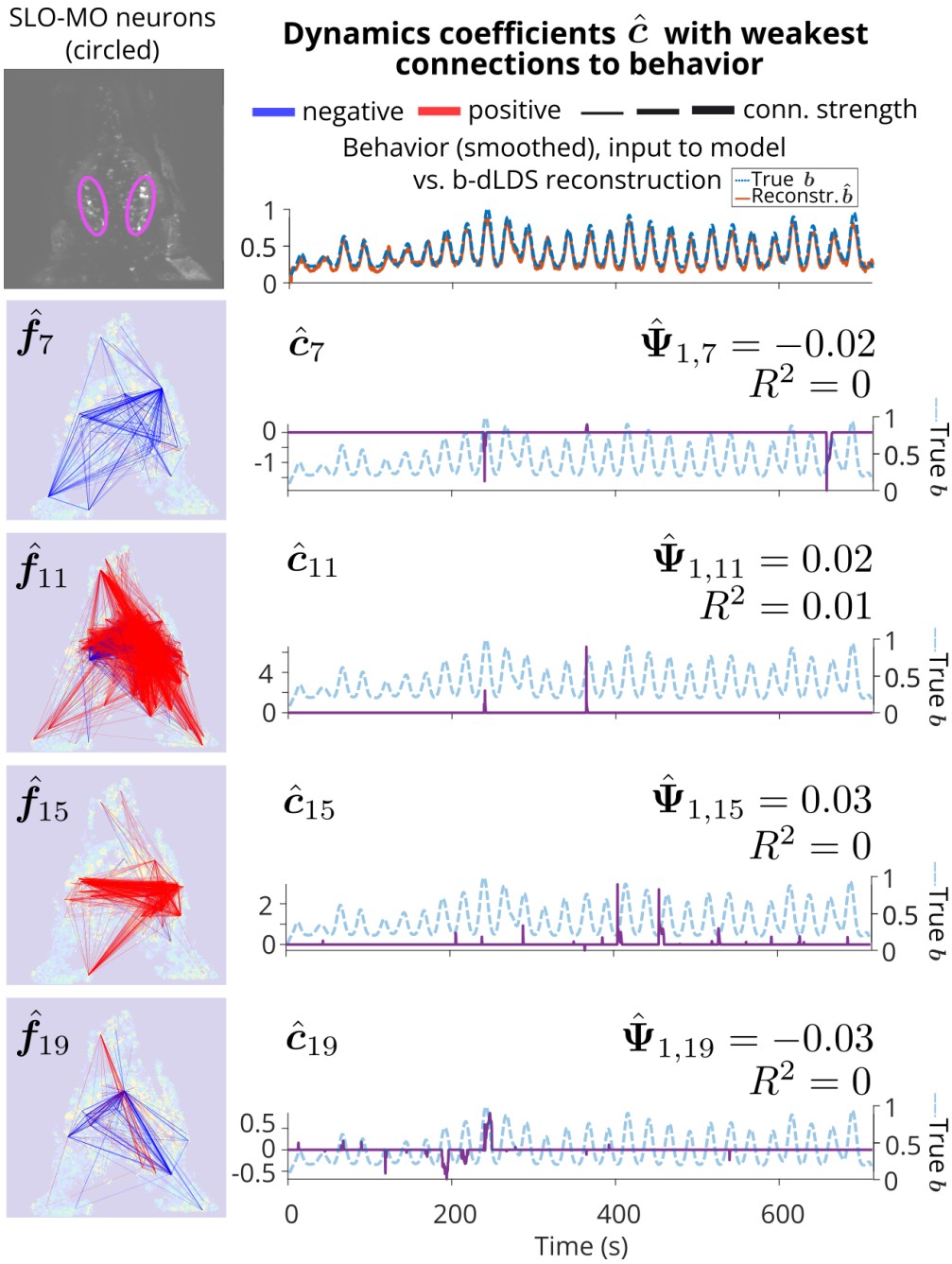

Supplementary Figure 6: **Four weakest $\Psi$ connections between dynamics coefficients and motor activity in a model with one behavior.** b-dLDS $R^2 = 0.91$ for neural data reconstruction (over whole time series concatenated; per neuron varies); $R^2 = 0.97$ for reconstruction of smoothed behavior data input to the model. Dynamic connectivity maps and dynamics coefficients shown. The weakest relationships to behavior are those dynamics operators that are rarely utilized throughout the recording.

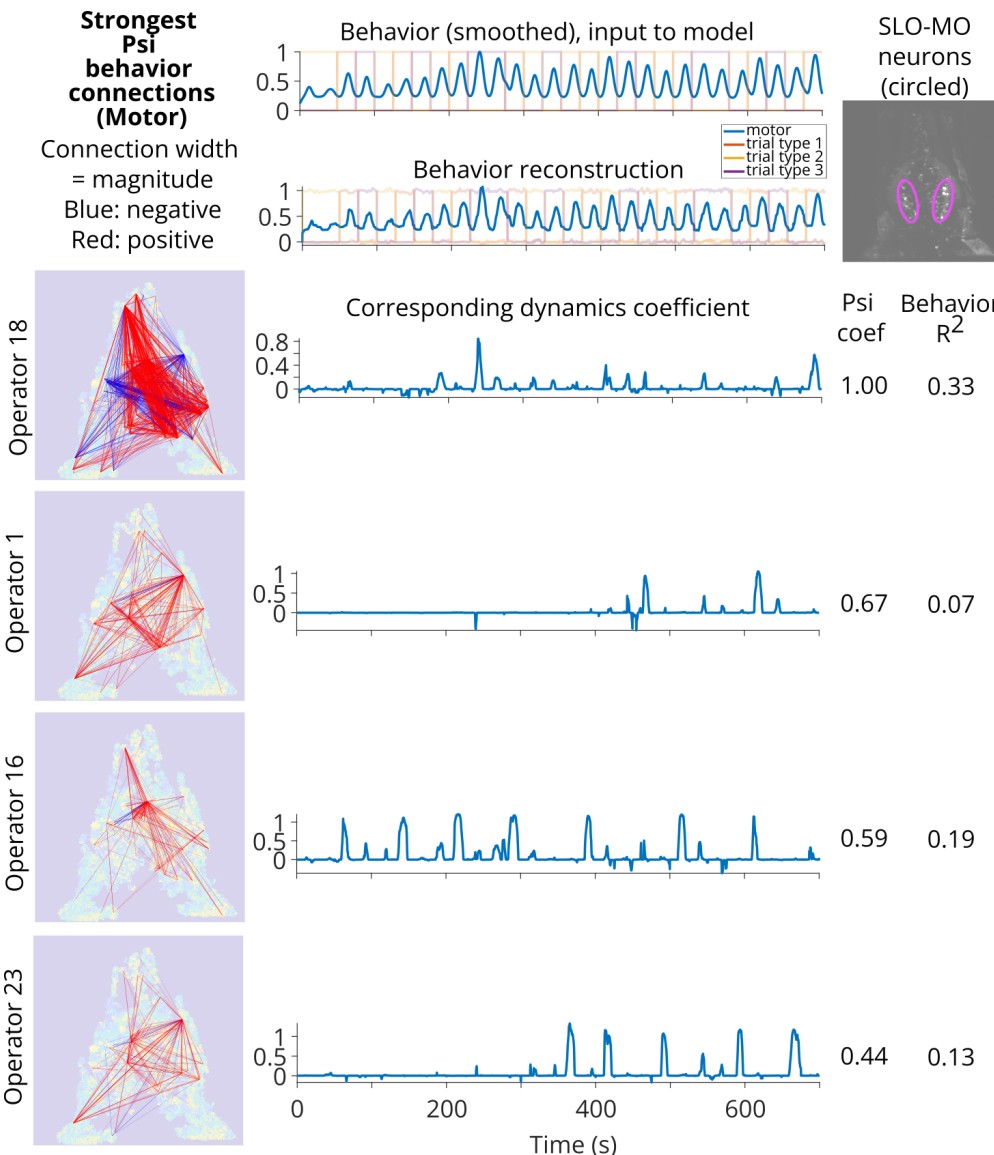

Supplementary Figure 7: **Four strongest $\Psi$ connections between dynamics coefficients and motor activity in a model with four behaviors (motor and 3 trial types based on stimulation patterns, i.e., fish offset by experimenter).** Overall model neural data reconstruction $R^2 = 0.93$; reconstruction of smoothed behavior data input to the model $R^2 = 0.997$ (all 4 behavior traces concatenated). Dynamic connectivity maps and dynamics coefficients shown. Motor-aligned dynamics are utilized throughout the recording or later in the recording.

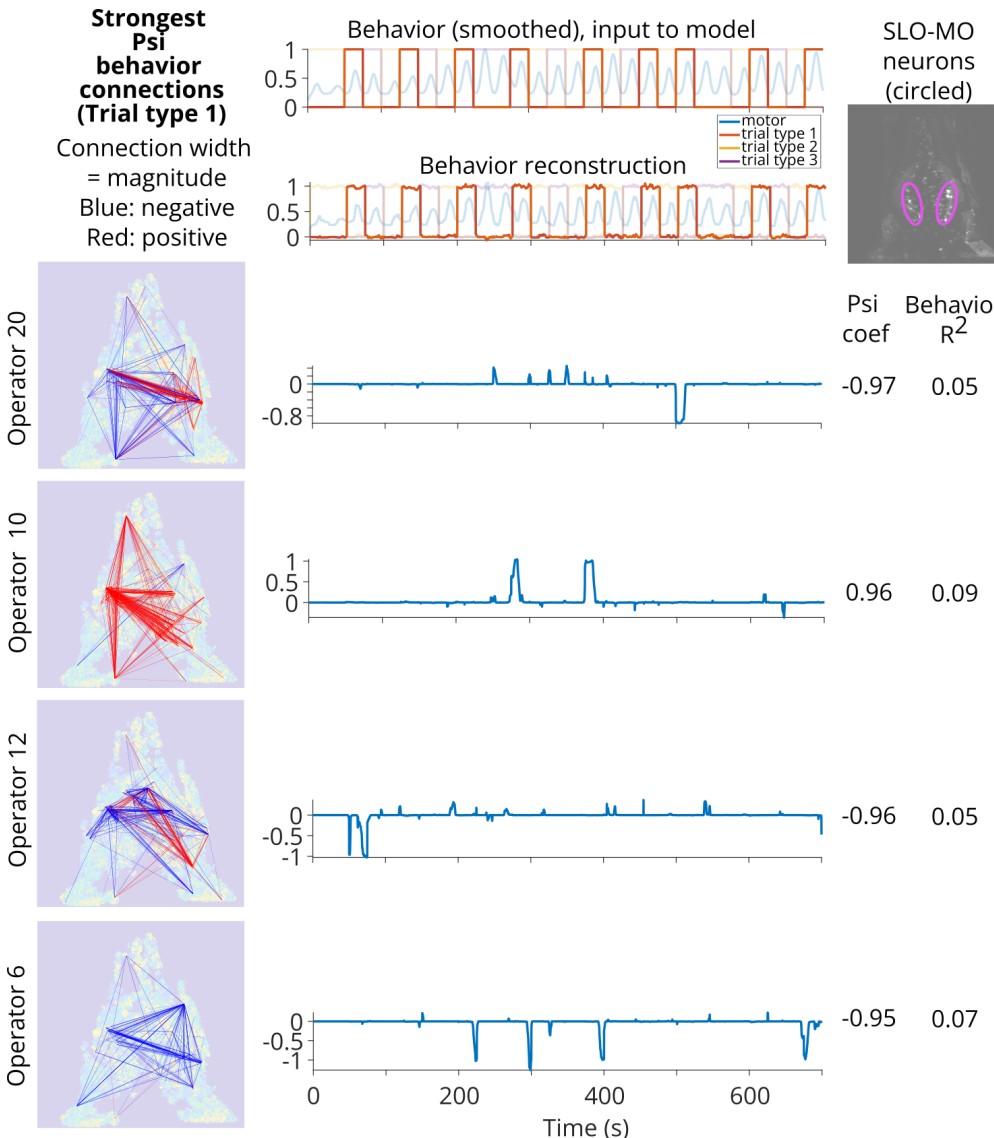

Supplementary Figure 8: **Four strongest $\Psi$ connections between dynamics coefficients and swim trial type 1 in a model with four behaviors (motor and 3 trial types based on stimulation patterns between swims).** Overall model neural data reconstruction $R^2 = 0.93$; reconstruction of smoothed behavior data input to the model $R^2 = 0.997$ (all 4 behavior traces concatenated). Dynamic connectivity maps and dynamics coefficients shown. Swim trial-aligned dynamics cover different time segments of the recording.

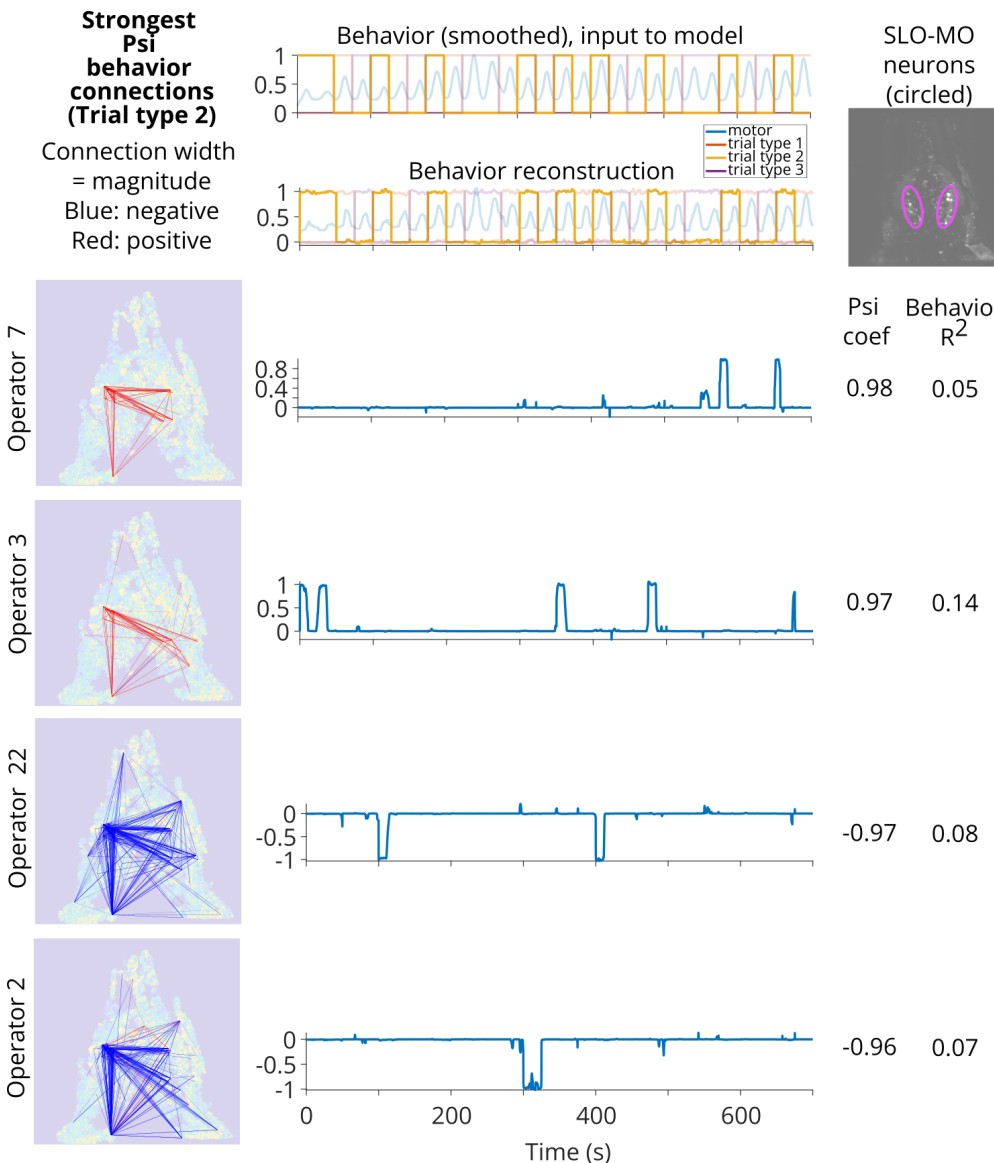

Supplementary Figure 9: **Four strongest $\Psi$ connections between dynamics coefficients and swim trial type 2 in a model with four behaviors (motor and 3 trial types based on stimulation patterns between swims).** Overall model neural data reconstruction $R^2 = 0.93$; reconstruction of smoothed behavior data input to the model $R^2 = 0.997$ (all 4 behavior traces concatenated). Dynamic connectivity maps and dynamics coefficients shown. Swim trial-aligned dynamics cover different time segments of the recording.

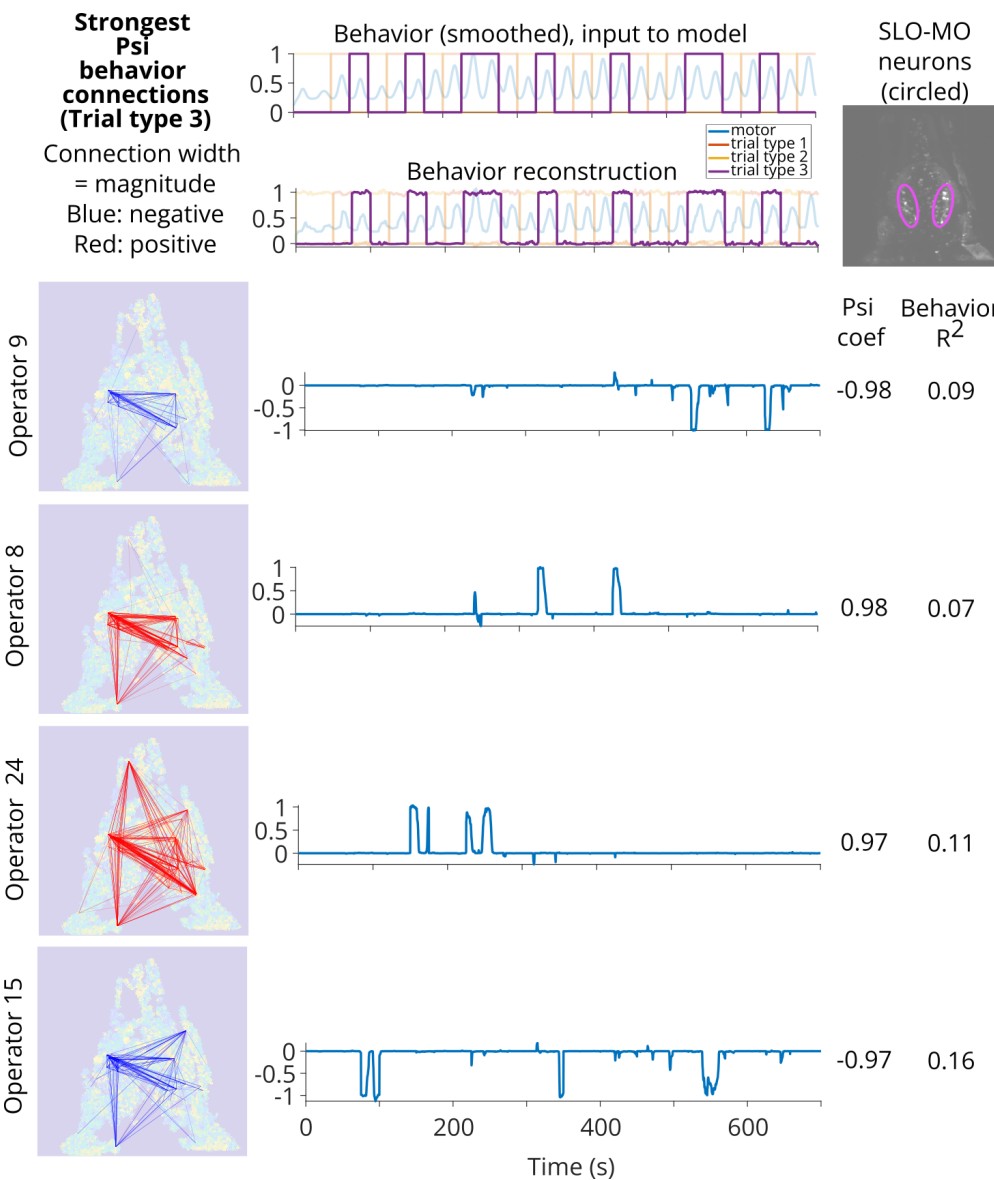

Supplementary Figure 10: **Four strongest $\Psi$ connections between dynamics coefficients and swim trial type 3 in a model with four behaviors (motor and 3 trial types based on stimulation patterns between swims).** Overall model neural data reconstruction $R^2 = 0.93$; reconstruction of smoothed behavior data input to the model $R^2 = 0.997$ (all 4 behavior traces concatenated). Dynamic connectivity maps and dynamics coefficients shown. Swim trial-aligned dynamics cover different time segments of the recording.

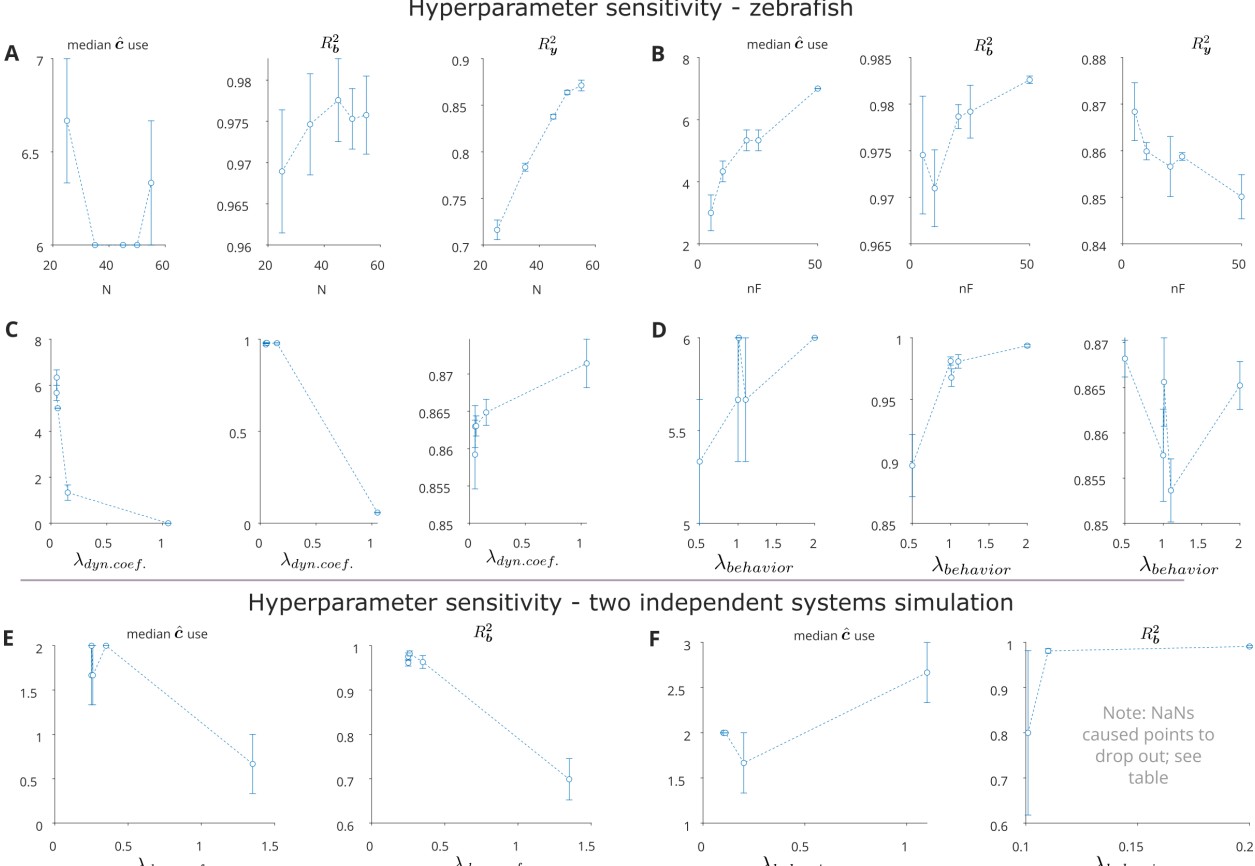

Supplementary Figure 11: **Hyperparameter sensitivity experiments related to number of dynamics coefficients and model size.** Data from Table 8. 3 seeds shown for each point. **A-D:** Zebrafish model parameter sweeps: N, nF, regularization for sparsity of $\widehat{c}$, regularization for $b$ reconstruction. **E,F:** Simulation (Fig. 3) parameter sweeps: regularization for sparsity of $\widehat{c}$, regularization for $b$ reconstruction.

