# OpenReview forum: "Behavior-dLDS: A decomposed linear dynamical systems model for neural activity partially constrained by behavior"
_TMLR — Accepted by TMLR_

### Review · Reviewer_v5yr · 2026-03-25

**Summary Of Contributions:**

This manuscript presents behavior-dLDS, which explains neural data in a low-dimensional latent space. The latent dynamics are represented with multiple dynamics operators and time-varying coefficients, yielding behavior from dynamics coefficients, not from latent space. Experiments on synthetic and zebrafish hindbrain calcium imaging datasets were performed to validate the behavior-dLDS.

**Audience:**

Yes

**Audience Explanation:**

Several researchers in specific fields, such as whole-brain literature, may find certain values in this study.

**Claims And Evidence:**

No

**Claims Explanation:**

I think the current version is not sufficient. Please see the Requested Changes below. Overall, I think the additional experiments on other datasets should be performed.

**Requested Changes:**

- Template is different from the standard TMLR format. Please make sure to use the correct template.
- The experiments were performed on synthetic and zebrafish recording datasets, which is still limited, while the claim is general. The authors should perform more experiments on other datasets.
- The behavior-dLDS requires extensive hyperparameters such as N, nF, sparsity regularization, and Frobenius penalty, whose tuning would require extensive hyperparameter search in practice. Readers might find it difficult to deploy it. Furthermore, the gain of the proposed method might come from the hyperparameter choice, rather than the advantage of the proposed method.
- Is it correct to add the Frobenius penalty (scalar) into the gradient (matrix) in Eq. 13? Furthermore, the nabla in Eqs. 12 and 13 should be with respect to \Psi, not D.
- The term E[x_t x_{t-1}^T] vanishes in Eq. 18, while Eq. 19 brings it again. The derivation from Eqs 15-21 should be revised carefully.
- Eq. 19 implicitly uses E[F x_{t-1} x_{t-1}^T] \approx E[F] E[x_{t-1} x_{t-1}^T], which is guaranteed by independence. However, the term F uses c_t from the time series, which breaks the independence, thereby leading to a contradiction in Eq. 19.
- The summation in Eq. 9 should be from l=1 to M, not from l=1 to L. After Eq. 13, m={1, …, M’} should be revised into m={1, …, M}.
- The term \tilde{F_t} should be defined explicitly.
- Check typos:
    - as with GLMS → as with GLMs
    - Rewrite the following: “to be statistically in terms of”
    - the minimum number of elements of c_t are → the minimum number of elements of c_t is
    - dirac-delta → Dirac-delta
    - 1 trials out of 40 → 1 trial out of 40
    - E: True dynamics coefficients → This should be C
    - PSIDencountered → PSID encountered
    - Equation equation 21 → Equation 21
    - to enable the fish maintain its position → to enable the fish to maintain its position
    - missing year in reference “Not everything, not everywhere, not all at once: a study of brain-wide encoding of movement.”

---

> ### Comment · Action_Editor_sVsK · 2026-04-02
>
> Dear reviewer,
>
> The authors have requested a two-week extension for submitting their rebuttal. Could you please wait for their response before making the final decision? Thank you for your contributions to TMLR!
>
> Best,
> AC

---

> > ### Comment · Reviewer_v5yr · 2026-04-03
> > **Sure!**
> >
> > Sure, no problem on my side. Thank you for kind words!

---

> ### Author Response · Authors · 2026-05-02
> **Review response - v5yr**
>
> We would like to thank Reviewer v5yr for highlighting opportunities for clarifying the rationale for our model, and for the copy edits. We hope that by presenting our results from further experiments, especially comparing b-dLDS to b-dLDS-x (mapping behavior to the latent states instead of the dynamics coefficients), we strengthen our case for the interpretability benefits of b-dLDS.
>
> 1. Template: Unfortunately we are a bit confused since we did use the TMLR LaTeX template, which shows the header “Under review as a submission to TMLR” on each page, etc. We of course want to make sure that we are using the correct template, so if there is any additional input the Reviewer or editor can give us as to a more correct template to use, we would like to correct this!
> 2. Additional experiments: We have now included additional experiments including: 1) Analysis of data from a task-driven RNN (RNN, based on the delayanti task in Driscoll et al. (2024)) and 2) ablation experiments on simulated data, and 3) testing of model variations on the  zebrafish data.
> The TaskRNN tests, in a controlled setting, how the c coefficients can capture the behavioral properties of the TaskRNN. This experiment, run in comparison with the equivalent version of our model where the behavior is directly related to the internal state activity (x instead of c), shows that the dynamics coefficients can better relate neuronal dynamics to external behavior. Specifically, this experiment reinforces the claim that b-dLDS models neural-behavior relationships well, and that mapping dynamics coefficients to behavior better allows the model to capture nonlinear neural-behavior relationships than would mapping to the latent states.
> The ablation experiments (Supplementary Figure 5 in the new version of the paper) are aimed at testing the contributions of the various modeling and fitting choices, including the \Psi update step size rescaling, the Frobenius norm regularization on \Psi, and the variability of the b-dLDS model across multiple seeds. We show that b-dLDS is robust across seeds and that the specifics of the step-size rescaling and regularization over the model fitting are necessary for good fits. This experiment reinforces the claim that the fitting procedure of b-dLDS is robust and requires the proposed modeling and fitting terms.
> We tested model variations on the zebrafish data (Table 3 in the new version of the paper), in particular to demonstrate the real-data utility of using c vs. x to relate neural activity to behavior. Specifically, we change in modeling the behavior as a function of the neural state x (which we call b-dLDS-x) vs. the b-dLDS model we propose with the behavior being a function of the dynamics coefficients c. We demonstrate on the zebrafish data that while b-dLDS and b-dLDS-x offer similar neural reconstruction and behavior reconstruction, b-dLDS still outperforms b-dLDS-x in terms of interpretability of the mapping \Psi. Specifically, b-dLDS more consistently shows that large \Psi values map to dynamics coefficients with a high R^2 with behavior; by contrast, the R^2 between b-dLDS-x latent states and behavior is smaller, even after optimizing the regularization parameter for better behavior reconstruction overall.
> These experiments reinforce that in some real data the coefficients c can be a better representation of the behavioral output of the animal than the latent neural representation x.
> We welcome any other recommendations that the reviewer has for experiments/datasets that can reinforce any of the other claims in the work.
> 3. Typos in text and equations: Thank you for pointing these out. We very much appreciate the attention to detail and have corrected the paper.

---

### Review · Reviewer_nmuo · 2026-03-26

**Summary Of Contributions:**

This paper introduces behavior-dLDS (b-dLDS), an extension of decomposed linear dynamical systems (dLDS) that adds a linear mapping from dynamics coefficients to observed behavior, regularized for sparsity. This allows the model to decompose neural dynamics into behavior-related and behavior-independent components. The authors validate b-dLDS on simulated data (recovering known dynamics-behavior mappings) and apply it to zebrafish hindbrain calcium imaging data, where they identify dynamic connectivity patterns associated with swimming behavior.

The idea of separating behavior-related from behavior-independent dynamics is well-motivated and addresses a real need in systems neuroscience. However, the paper's central modeling assumption is not empirically tested, the simulations are circular (testing the fitting procedure on model-generated data), and the neural data application is too thin to evaluate whether the approach delivers on its promise. In its current form, the paper does not provide sufficient evidence that b-dLDS produces a meaningful decomposition of real neural data.

Strengths and Weaknesses

Strengths
- The problem is well-motivated: large-scale neural recordings contain both behavior-related and behavior-independent activity, and existing models either ignore behavior entirely (dLDS) or condition all dynamics on behavior (CLDS). The "middle ground" positioning is sensible and clearly articulated.
- The sparse regularization on psi is an elegant solution that naturally identifies which dynamics operators relate to behavior. The Frobenius norm option for pushing columns of psi toward zero is a nice touch for low-dimensional behavior.
- The comparison with CLDS and PSID (Fig. 3) clearly illustrates the structural difference between the approaches: b-dLDS learns the block-diagonal dynamics structure while CLDS and PSID cannot. The scalability to 13,000 neurons is a practical advantage over CLDS and PSID.

Weaknesses
The main weaknesses fall into three categories: an untested core assumption, insufficient model validation, and an underdeveloped neural data application.

1. The core modeling assumption (behavior generated by dynamics coefficients rather than latent states) is not justified or tested. The central novelty of b-dLDS is Eq. 4, mapping behavior from dynamics coefficients rather than latent states. The authors motivate this with one sentence about x_t fluctuating too quickly (p.4), but this is asserted rather than demonstrated. Other models (PSID, CEBRA) map behavior from latent states and work well. Moreover, the dynamics coefficients in the model are doing double duty (regulating state transitions and serving as the behavioral readout). This is a strong assumption and if it doesn't hold, the decomposition into behavior-related and behavior-independent DOs may be artifactual.

2. The simulations test the fitting procedure, not the model's validity as a hypothesis. All four simulation cases generate data from the b-dLDS generative model itself, so R squared around 1 is expected. These simulations verify that the optimization procedure works, but do not test whether the b-dLDS decomposition captures meaningful structure in data where the generative process differs from the model (e.g., an RNN). On a related note, the paper claims a "conceptual reframing of the relationship between brain activity and behavior" (p.10). This suggests that the authors see their model as more than a data analysis tool, but as a conceptual model for the brain. For that stronger claim to be supported, the goodness of fit of the model on brain data needs to be compared to different proposed generative models.

3. The zebrafish application does not evaluate model fit and provides minimal scientific interpretation. The paper claims b-dLDS "can accurately model 13000-dimensional zebrafish hindbrain calcium imaging data" (p.10), but neural reconstruction accuracy is never reported. The only R squared (0.90, Fig. 4 legend) is labeled "Overall model" without clarification of what it covers (behavior reconstruction of a 1D motor signal is a much easier problem than reconstructing 13,000-dimensional calcium dynamics). The dynamic connectivity analysis (the primary use case) is only loosely motivated, in Appendix A. The interpretation is almost entirely descriptive ("these patterns may indicate nonstationarity," p.9) without connecting to known zebrafish hindbrain biology or explaining what the model reveals beyond Yang et al. (2022).

**Audience:**

Yes

**Audience Explanation:**

This paper will be of interest to the neuro-AI audience, and researchers using ML tools for the joint analysis of neural and behavioral data.

**Broader Impact Concerns:**

No concerns. The paper presents a method for analyzing neural recordings and does not raise ethical issues beyond standard considerations for neuroscience research.

**Claims And Evidence:**

No

**Claims Explanation:**

The paper's central claims are not adequately supported by the evidence presented. (1) The core modeling assumption that behavior is generated by dynamics coefficients rather than latent states is motivated by a single assertion but never tested empirically against the alternative. (2) All simulations generate data from the b-dLDS model itself, so the high reconstruction accuracy is expected and does not validate the model as a hypothesis about brain-behavior relationships. (3) The claim that b-dLDS "can accurately model 13000-dimensional zebrafish hindbrain calcium imaging data" is not supported: neural reconstruction accuracy is never reported, the "Overall model R = 0.90" is ambiguous, and the scientific interpretation of results is minimal. The well-motivated problem and elegant regularization approach could support the claims with additional validation, but the current evidence is insufficient.

**Requested Changes:**

Critical for acceptance

1. Test the c→b assumption empirically. Compare b_t = Ψc_t against b_t = Ψx_t (or another latent-state-based mapping) on both zebrafish data and simulated data from a task-trained RNN (not a generative model that is explicitly designed to match b-dLDS). If the c_t mapping is better, show it. If not, acknowledge the choice as a modeling convenience and adjust claims accordingly.

2. Include at least one simulation with a misspecified generative model. Generate data from a process that does not conform to b-dLDS assumptions (e.g., an RNN trained to perform a cognitive or behavioral task) and evaluate whether the learned decomposition recovers meaningful structure. This is essential for distinguishing "b-dLDS captures real structure" from "b-dLDS imposes structure via regularization."

3. Report and break down model fit metrics for the zebrafish data. Clarify what "Overall model R = 0.90" measures. Report neural reconstruction R separately from behavior R. Discuss in the main text, not just in figure legends. Ideally, model comparison goodness of fit/reconstruction accuracy between different models.

4. Provide scientific interpretation of the zebrafish results. Motivate the dynamic connectivity analysis in the main text. Connect the behavior-related vs. behavior-independent DOs to known zebrafish hindbrain biology. State explicitly what new insight the model provides beyond Yang et al. (2022). If possible, the work would be greatly strengthened if the model were applied to a dataset with more complicated behavior.


Would strengthen the work

5. The CLDS comparison (Fig. 3) should use a more balanced simulation. The current simulation is designed with independent subsystems that CLDS cannot represent by construction. A comparison on data with partially overlapping behavior-related and behavior-independent dynamics would be more informative. Also compare reconstruction accuracy on the zebrafish data.

6. Clearly delineate what is new in b-dLDS vs. inherited from dLDS. The paper presents the full dLDS framework (Sections 2–3) before revealing that the new contribution is Ψ and the associated regularization. This should be made explicit early in Section 3. The "conceptual reframing" language in the Discussion (p.10) overstates what is effectively a regularization extension; the claims should be calibrated accordingly.

7. Clarify the interpretive roles of x_t vs. c_t. The model infers two sets of variables, both shown in figures, but the paper provides no guidance on when a practitioner should examine one vs. the other. Since behavior is read out from c_t, what is x_t useful for in b-dLDS? A short paragraph in the Discussion would help.

8. Compare against dLDS + post-hoc behavior correlation. This is a natural baseline. Run unsupervised dLDS, then correlate learned c_t with behavior. If post-hoc correlation gives a similar decomposition, the behavior regularization is not adding much.

9. Discuss hyperparameter sensitivity. Tables 1–2 list 15+ hyperparameters, and the supplement notes that R is "particularly sensitive to model size" (B.4.2). Since regularization directly determines how many DOs are classified as behavior-related, the authors should show (or at least discuss) how robust the decomposition is to reasonable parameter variation.

10. Trial types are not behavior, and are not well-defined. The zebrafish analysis with trial type information (Supp. Figs. 6–9) treats experimenter-defined conditions as "behavior" input. The authors should acknowledge this distinction and define the trial types in the main text. Furthermore, what are the different types of trials? I did not find this in the text.

11. Discuss n_F selection in the main text. The model uses 15 DOs for data with 6 ground-truth operators (Fig. 2). The heuristic for choosing the number of dynamics factors is buried in Table 1 of the Appendix. Since over-parameterization affects interpretability, this should be discussed in the main text.

12. Clarify the status of noise variances. The noise variances are not in the parameter set (Section 4, p.5).

13. Include a code and data availability statement. Given the large number of hyperparameters, a reference implementation is important for reproducibility.

14. Typo, p.3: "Sparsity in this case encourages the learned basis to be statistically in terms of when they are active". Missing word after "statistically" (likely "independent").

---

> ### Comment · Action_Editor_sVsK · 2026-04-02
>
> Dear reviewer,
>
> The authors have requested a two-week extension for submitting their rebuttal. Could you please wait for their response before making the final decision? Thank you for your contributions to TMLR!
>
> Best,
> AC

---

> > ### Comment · Reviewer_nmuo · 2026-04-02
> > **extension ok**
> >
> > yup  - fine by me!

---

> ### Author Response · Authors · 2026-05-02
> **Review response - nmuo**
>
> We would like to thank Reviewer nmuo for their detailed guidance on experiments that would test the central assumptions of the model and additions to the text that would strengthen the paper overall.
>
> Critical for acceptance
>
> 1. Test the c→b assumption empirically. We thank the Reviewer for this recommendation. This is important to test and central to our modeling choice. We thus have implemented and tested a version of b-dLDS that maps the behavior to the latent states x instead of the dynamics coefficients c (which we call b-dLDS-x). We have  tested b-dLDS against b-dLDS on both the original zebrafish example included in the original manuscript, and an additional task-trained RNN dataset (more below in (2), misspecified generative model) with a nonlinear relationship between activations and behavior. These tests reveal that the difference between b-dLDS and b-dLDS-x greatly impacts the interpretability of the model in terms of connecting dynamical interactions within the neural population with behavioral epochs.  This impact to interpretability manifests in both the sparsity of the mapping Psi to behavior, and the consistency of the model dynamics over behavioral epochs.
>
> 	For the RNN example, we added Figure 4 and Table 1, which show that, as we increase the regularization in b-dLDS to make Psi sparser, it becomes striated in the vertical direction, i.e., a small number of dynamics coefficients map explain  the behavior traces. Conversely, in the b-dLDS-x case, Psi remains dense in order to continue reconstructing the behavioral traces with high fidelity, i.e., Psi must use a linear combination of all of the latent states x in order to reconstruct the behavior, no matter how strongly constrained the model is to bias it against that solution. Moreover, in b-dLDS the same linear dynamical system accurately reconstructs the internal RNN activations for each behavioral epoch of the task, while b-dLDS-x alternates between many systems in order to maintain a consistent external behavioral readout despite the RNN state changing on a faster time-scale. This emphasizes our main point that more quickly changing internal (neural) dynamics can occur during slower-scale behavior, and that using the dynamics rather than the latent state to connect neural activity and behavior can offer a more interpretable model.
>
> 	In the zebrafish comparison, we sought to quantify a similar question of concentration of behavioral variance explained throughout the c (for b-dLDS) or x (b-dLDS-x) by looking at whether the largest values in the Phi matrix corresponded to the latent variable (x or c) that had the highest R^2 in reconstructing the behavior. We found that for b-dLDS, the strongest Psi value had the same index as the dynamics coefficient with the greatest R^2 with behavior in 7/9 cases. For comparison, out of the different parameter sets tested for b-dLDS-x, the strongest Psi value had the same index as the latent state with the greatest R^2 with behavior in only 4/15 cases. Moreover, the values of the max c R^2 with behavior were 0.69-0.95 for b-dLDS, whereas the values of the max x R^2 with behavior were 0.26-0.45, indicating that the c’s are better able to concentrate explanatory power in terms of the behavior over the x’s.
>
> 	Said another way, in real, large-scale neural data, the c’s in b-dLDS give better interpretability by finding a more concise way of representing behavior-related dynamics than what the latent state provides.
>
> 2. Include at least one simulation with a misspecified generative model. We thank the Reviewer for their recommendation of a specific class of synthetic dataset that it would be helpful to see tested with b-dLDS. We applied b-dLDS and b-dLDS-x (as above) to the delayanti task data from Driscoll et al. (2024), which trains an RNN to perform multiple behavior tasks similar to a reaching task, with input coordinates on a circle as well as a task rule, and corresponding output coordinates as the behavior (in the case of the delayanti task, the output coordinates should be 180 degrees from the input coordinates). The input and output traces turn on and stay on for 25 time steps. Meanwhile, the RNN activations that capture this input, hold it in memory during the delay period, and then output the correct coordinates, vary smoothly and primarily around key transition points. Thus there is a nonlinear relationship between the activations and the behavior. We present the results on this model using b-dLDS and b-dLDS-x in Figure 4 and Table 1 to emphasize that in this dataset, which is not drawn from the generative model, modeling the input/output behavioral traces as being related to the dynamics coefficients c provide a more interpretable model than using the latent state x. We very much appreciate this suggestion as we feel that this experiment emphasizes the core modeling claim of the paper.
>
> (continued)

---

> > ### Author Response · Authors · 2026-05-02
> > **Review response - nmuo - part 2**
> >
> > (continued)
> >
> > 3. Report and break down model fit metrics for the zebrafish data. We thank the Reviewer for their recommendation. We agree that this was not clear, and we have added this to the paper text and captions (Figure 5, Supplementary Figures 6-10).
> >
> > 4. Provide scientific interpretation of the zebrafish results. We have added to and focused our zebrafish results on asymmetry in the dynamic connectivity, specifically in showing that there is asymmetry in the hindbrain activity networks that goes beyond imaging/geometric asymmetry. This example builds on past literature that has identified  asymmetry in other zebrafish brain regions, but has not been reported previously in the hindbrain. We quantified this effect, which builds on the ability of b-dLDS to analyze large-scale data and forms a hypothesis that can be followed up on by designing targeted future experiments.
> >
> >
> > Would strengthen the work
> >
> > 5. The CLDS comparison (Fig. 3) should use a more balanced simulation. We find CLDS to be a compelling and powerful model. What we aim to show in the simulations we presented is the importance of being able to decompose dynamics when they do exist in the data. When they are not relevant, CLDS has good performance, as presented in their paper. Moreover, it is quick and relatively easy to tune and run, and it is well-documented. We have now included another synthetic dataset based on an RNN (misspecified generative model), but we did not test CLDS on this or the zebrafish data because there are more than 10 behavior traces in the RNN data and there are too many neurons in the zebrafish data. CLDS unfortunately cannot run on such large data due to memory and computational limitations.
> >
> > 6. Clearly delineate what is new in b-dLDS vs. inherited from dLDS. Thank you for bringing to our attention this point for clarification. We have added text to the “behavior-dLDS model” and “Model inference” sections to clarify the technical innovations of b-dLDS. The “conceptual reframing” to which we refer in the discussion, however, is more focused on how we link behavior to the coefficients in the latent dynamics rather than most current literature, which models behavior as being a function of the latent state. We feel that this formulation is unique among models that explicitly model neural activity and behavior side by side, and our additional comparisons to b-dLDS-x (see above) we feel emphasizes the benefit of this model on interpretability.
> >
> > 7. Clarify the interpretive roles of x_t vs. c_t. We thank the Reviewer for highlighting this point. While we do not focus on x_t in our paper, there are possible directions for interpreting x_t that we have added to a point in the Discussion. Moreover we have added the b-dLDS-x model comparison to further demonstrate the practical difference in behavior modeling.
> >
> > 8. Compare against dLDS + post-hoc behavior correlation. We have now included comparisons to dLDS  in terms of maximum R^2 between dynamics coefficients and behavior, as well as between latent states and behavior, for dLDS, b-dLDS, and b-dLDS-x as described above - please see Appendix Section “What if b-dLDS mapped behavior to the latent states?” and Table 3. R^2 between any dynamics coefficient or latent state and behavior never exceeded 0.36 for dLDS.
> >
> > 9. Discuss hyperparameter sensitivity. The Reviewer rightly points out that regularization dictates how many dynamics coefficients are inferred as active and found to be behavior-related, which is important for the interpretation of the results. For the simulations where the ground truth is known, we have added experiments to show the impact of small perturbations in the regularization on the sparsity of c on behavior reconstruction and median number of active dynamics coefficients (Table 8, Supplementary Figure 11).
> >
> > 10. Trial types are not behavior, and are not well-defined. Thank you for bringing this to our attention. This is an interesting point since different studies have variable assessments/categorizations of behavior. Some use direct physical body positions, e.g., through video keypoint identification. Others use manual annotation of trials/bouts, e.g., studies of social behavior or decision making. We clarify in the introduction how behavior can take a number of forms in the introduction to better situate how we are considering the range of neuroscientific studies. We thus have, in b-dLDS, allowed the model to take in as “behavior” all types of inputs or outputs that align in time with the neural activity (e.g., as in the task-RNN example we have added). This may include behavior or other environmental time trace information. In the case of the zebrafish, the trial types are a type of behavior-related information describing when each type of displacement trial is occurring throughout the recording session (a sort of one-hot encoding). We have added text to Appendix Section “Trial types” to clarify how these were defined experimentally.
> >
> > (continued)

---

> > > ### Author Response · Authors · 2026-05-02
> > > **Review response - nmuo - part 3**
> > >
> > > (continued)
> > >
> > > 11. Discuss n_F selection in the main text. We have added text on this point to each Results subsection where each experiment is introduced, as well as a sensitivity analysis in Table 8 and Supplementary Figure 11.
> > >
> > > 12. Clarify the status of noise variances. We have now included this in the text of Appendix Section “Simulation of two independent systems and behavior” (since it is not a model parameter but rather a feature of one of the simulated datasets).
> > >
> > > 13. Include a code and data availability statement. We have now included these in a general format and will fill in the details when the double-blind review process is complete.
> > >
> > > 14. Typo, p.3. Thank you for noticing this typo; we have corrected it.

---

> > > > ### Comment · Action_Editor_sVsK · 2026-05-09
> > > >
> > > > Dear Reviewer,
> > > >
> > > > The authors have submitted their rebuttal. Please proceed with the discussion and finalize your recommendations. If any additional clarification is required, feel free to raise your questions to the authors.
> > > >
> > > > Thank you for your contributions to TMLR.
> > > >
> > > > Best, AE

---

### Review · Reviewer_arRv · 2026-03-26

**Summary Of Contributions:**

This paper introduces behavior-dLDS (b-dLDS), a behavior-aware extension of decomposed linear dynamical systems for large-scale neural recordings. The main idea is to keep the latent neural state model and time-varying linear dynamics decomposition of dLDS, but add a separate linear map from the dynamics coefficients $c_t$ to observed behavior $b_t$, rather than forcing behavior to directly determine all dynamics. This is intended to place the method between fully unsupervised dLDS and strongly behavior-conditioned models such as CLDS. Some latent dynamics can be tied to behavior, while others can capture parallel internal computations. The model is optimized with an EM-style procedure, where the E-step jointly infers latent states and dynamics coefficients under reconstruction, smoothness, sparsity, and behavior-matching terms, and the M-step updates the observation matrix, dynamics operators, and behavior map.

Empirically, the paper presents two types of evidence. First, on controlled simulations with known ground-truth dynamics and behavior mappings, b-dLDS is reported to recover the correct subset of dynamics coefficients tied to behavior and to reconstruct dynamics more accurately than CLDS, including across multiple settings with 10 random seeds. Second, the paper applies b-dLDS to a zebrafish hindbrain recording with roughly 13,000 neurons and one-dimensional motor behavior, using the learned $\Psi$ map to distinguish behavior-related from non-behavior-related dynamics and to visualize behavior-aligned dynamic connectivity patterns.

The main strengths are that the paper addresses a genuinely important neuroscience modeling problem, proposes a conceptually clean middle ground between unsupervised and behavior-conditioned latent dynamics models, and demonstrates scalability to a much larger neural dataset than the simulation benchmarks.

The main weaknesses are that the evidence is concentrated in a narrow simulation setup plus a single real biological dataset, several comparisons depend on simulation assumptions favorable to the model (for example $D=I$ in the simulations), and the real-data conclusions remain partly interpretive because there is no direct ground truth for which learned dynamics are truly behavior-generating versus internally driven.

**Audience:**

Yes

**Audience Explanation:**

I expect meaningful interest from at least three communities, computational neuroscience researchers working on neural population dynamics, machine learning researchers developing latent dynamical systems for partially observed high-dimensional time series, and scientists analyzing large-scale brain-wide recordings with behavioral annotations.

**Broader Impact Concerns:**

The paper is not an obvious source of direct societal harm in the way that some dual-use generative or surveillance models are, and the manuscript does include a limitations section. However, I do not see a dedicated Broader Impact discussion in the visible text. A brief statement would still be useful.

**Claims And Evidence:**

Yes

**Claims Explanation:**

The central modeling claim, "b-dLDS can provide a middle ground between behavior-conditioned and behavior-agnostic decomposed dynamical systems, and can separate behavior-related from non-behavior-related dynamics better than a fully behavior-conditioned comparator," is supported reasonably well in the controlled simulations. The evidence is less definitive for the broader biological interpretation claims on the zebrafish data, where the method appears promising and plausible but not fully validated.

**Requested Changes:**

### Critical:
1. The paper should provide a more explicit discussion of identifiability and ambiguity in the decomposition. Because the method separates behavior-related from non-behavior-related dynamics through regularization and a learned $\Psi$ map, readers need a clearer account of when this separation is unique, when it is only one plausible decomposition, and how sensitive it is to regularization strength.

2. The real-data claims should be toned down or validated more directly. The zebrafish section is interesting, but most of the strongest biological conclusions remain interpretive. The paper would be more convincing if it either reduced the scope of its biological claims or added stronger external validation analyses.

3. The comparison to CLDS and PSID should be made more carefully comparable. The manuscript notes that CLDS requires an 80-20 split and was trained on 40-50 trials in the simulation comparison, and that PSID is disadvantaged by its shared latent-state assumption in the constructed simulation. That may be fair, but the paper should clearly acknowledge that the benchmark is aligned to highlight the conceptual distinction rather than to serve as a fully general head-to-head comparison.

### Would strengthen the work:

1. Add stronger ablation studies on regularization and model size, especially for $\lambda_4$ (behavior reconstruction), sparsity on $c_t$, and the sparsity or Frobenius options used for $\Psi$.

2. Add a more quantitative evaluation of the zebrafish dynamic connectivity maps, beyond visual interpretation and biological consistency arguments.

3. Include runtime or memory benchmarks for b-dLDS itself, not only the statement that CLDS and PSID encountered memory and size errors on the zebrafish dataset.

---

> ### Comment · Action_Editor_sVsK · 2026-04-02
>
> Dear reviewer,
>
> The authors have requested a two-week extension for submitting their rebuttal. Could you please wait for their response before making the final decision? Thank you for your contributions to TMLR!
>
> Best,
> AC

---

> > ### Comment · Reviewer_arRv · 2026-04-03
> >
> > I am good with it.

---

> ### Author Response · Authors · 2026-05-02
> **Review response - arRv**
>
> We thank Reviewer arRv for highlighting the plausibility and utility of our model for neuroscience data, as well as for identifying several points of confusion about our experimental design and opportunities for improving the validation. We have added ablation experiments, quantitative analyses, and text that we hope will clarify the strengths of our model and possibilities for future exploration of neural connectivity asymmetry in the zebrafish.
>
> Critical:
>
> 1. Identifiability and ambiguity in the decomposition: We thank the reviewer for raising this important point, and we have added text to the Discussion on this topic. Identifiability is undoubtedly an open question with dLDS-family models. We do not currently know of a theoretical guarantee, especially for a large-scale dataset such as our neuroscience application, that shows that the decomposition presented by a particular instance of b-dLDS (or previous models in this space, e.g., dLDS, CLDS, etc.) is the best or only possible decomposition. Future work in this direction would rely on theory from dictionary learning (Wu et al. 2018, Cohen et al. 2019, Hu et al. 2023). We note that for dictionary learning, the theoretical bounds were developed after the methods were proposed and empirically tested under many conditions to form a foundation of intuition for the model, so we hope that future efforts will also clarify the sample or model size or parameter requirements for identifiability for dLDS-class models as well.
>
> 	That said, there are currently multiple ways to build confidence in the consistency of downstream analyses. For example, in the dLDS paper, Mudrik et al. (2024) showed that dLDS is robust to initialization. We now also show for b-dLDS multiple model comparisons across random seeds (Table 3 in the revised paper) and compare features of the outputs (e.g., coefficients, dynamic connectivity maps). We have also included these and a sensitivity analysis in Table 8 and Supplementary Figure 11 to also test consistency across model size and key regularization parameters.
>
> 2. The real-data claims should be toned down or validated more directly: We agree, and in our revision we have focused our real-data claims to: (1) center our biological claims on the more narrow question of asymmetric networks identified by the dynamic connectivity learned in the b-dLDS model, and (2) demonstrate the robustness of b-dLDS’s improved interpretability via model comparisons on the zebrafish data (Table 3), complementing similar ablation studies on the synthetic data. With respect to asymmetry, we have quantified our observation (Results Section “b-dLDS highlights behavior-related dynamic connectivity and asymmetry across the zebrafish hindbrain”, Table 2), validating the finding of the asymmetric nature of the identified dynamic operators. We ground these results in the literature, in which meaningful asymmetry has been reported in the zebrafish brain, but whether and how it presents in the hindbrain is an open question. We thus demonstrate that b-dLDS can help propose asymmetric neural connectivity of interest that can be followed up on in the design of new in vivo experiments. We have edited the text to highlight the complexity and relevance of this particular aspect of zebrafish neural activity and to better describe the behavior dataset as well.
>
> 3. The comparison to CLDS and PSID should be made more carefully comparable: We thank the reviewer for noting these points for further clarification and have restated the way we did comparable train/test sampling for each of these models, as well as the conceptual reasoning for comparing all 3 models on synthetic data generated from independent, simultaneously active ground truth systems. Specifically, these simulations were chosen for their relevance to neuroscience because whole brain recordings may feature such independent but concurrent systems.
>
> 	However, the key conceptual question here is that of comparing the dynamics coefficient-behavior mapping to the latent states-behavior mapping. To that end, we have now derived, implemented, and tested b-dLDS-x, where the behavior is modeled by a linear readout (through Psi) of the latent vector x instead of the dynamics coefficients c. We show results of this comparison between b-dLDS and b-dLDS-x on the zebrafish data (Table 3), as well as another synthetic dataset, which we call TaskRNN, based on the delayanti task in Driscoll et al. (2024). Note: in the new TaskRNN dataset, there are 20 input traces and 3 output traces, all of which we input to the model as “behavior” traces, and which are too high-dimensional to be analyzed with CLDS. Instead we use this example to test how b-dLDS operates on a simpler, interpretable dataset that is not drawn from the model, and to test the validity of modeling the behavior through the dynamics coefficients c (i.e., instead of a readout of x).
>
> (continued)

---

> ### Author Response · Authors · 2026-05-02
> **Review response - arRv - part 2**
>
> (continued)
>
> On this new dataset, we demonstrate that b-dLDS can reconstruct the RNN’s activations (Figure 4, Table 1), which increase and decrease continuously, and behavior traces, which turn on, stay constant, and then turn off. Moreover, we compare b-dLDS to b-dLDS-x and show that, in the case of this benchmark dataset featuring a nonlinear relationship between activations and behavior, b-dLDS can more naturally model the link between the input/output behavior and the internal dynamics. We measure this through the sparsity of the readout Psi where a small number of dynamics coefficients map to the behavior in b-dLDS, but b-dLDS-x cannot achieve the same type of sparsity in terms of the latent states x.
>
>
> Would strengthen the work:
>
> 1. Add stronger ablation studies on regularization and model size: We thank the Reviewer for their recommendation. We have now run ablation studies on the step size rescaling and Frobenius norm on \Psi based on simulated data (Supplementary Figure 5), and also show a comparison across models on zebrafish with and without this regularization on \Psi (Table 3). Ablation of behavior reconstruction is just dLDS, so we included a comparison to dLDS in Table 3 to show that post hoc correlation between dynamics coefficients and behavior is much weaker and less interpretable than with b-dLDS. We note that the sparsity on c_t was previously explored in the original dLDS paper (especially in the FitzHugh Nagumo experiments) and shown to improve interpretability, as well.
>
> 2. Add a more quantitative evaluation of the zebrafish dynamic connectivity maps: We have now quantified baseline and dynamic connectivity map asymmetry, as described above.
>
> 3. Include runtime or memory benchmarks for b-dLDS itself: We have now included these in Table 3, with the caveats that (1) these models were run on a shared server and runtimes were subject to the effects of other processes running on the servers, and (2) there is no built-in Linux memory tallying function for MATLAB, so the values presented are in the ballpark at best, and at worst they are negative because they were generated by subtracting memory usage at the start from the memory usage at the end of each experiment. We would be open discussing with the editor/reviewer if these should be included given the uncertainty (specifically for the RAM usage).

---

### Author Response · Authors · 2026-05-02
**Reviewer response summary**

We would like to thank the reviewer and editors for their time spent reviewing our paper, their helpful recommendations, and their flexibility. In addition to editing the text for clarity on a number of conceptual and theoretical points, we have focused our zebrafish analysis on asymmetry in connectivity, added experiments on a new synthetic dataset (Driscoll et al. 2024), run ablation and sensitivity experiments on several key parameters with different datasets, and compared b-dLDS to b-dLDS-x, a new version of b-dLDS that maps latent states to behavior, in order to test the dynamics coefficients to behavior mapping assumption central to the technical innovations of our model. We hope that the reviewers will find the new version of the paper to be a more holistic and detailed exploration of the interpretability benefits of b-dLDS.

We would like to especially like to  note the following changes:

1. Testing our central dynamics coefficients-to-behavior assumption via b-dLDS-x: We implemented and tested b-dLDS-x on both biological and synthetic data. By analyzing the resulting mapping Psi and correlation between latent states and behavior, we found that b-dLDS still provides a sparser, more interpretable mapping, with more of the behavior correlation concentrated in and strongly mapped to a small number of dynamics coefficients. We also showed that post hoc correlation with behavior using the original dLDS model is not nearly as good at finding behavior-correlated dynamics by chance.

2. Biological findings from zebrafish data - quantifying asymmetry: We have focused our zebrafish analysis on asymmetry in connectivity in the hindbrain. This was noted in our figures by several colleagues who work with zebrafish on a day-to-day basis. This is interesting because while asymmetry has previously been identified in brain areas such as the habenulae in the forebrain, as well as in behavior, hindbrain asymmetry remains underexplored and could be a place for hypothesis generation and further in vivo experimentation.

3. New synthetic dataset - task-driven RNN: We tested b-dLDS and b-dLDS-x on synthetic data from an RNN performing a delayanti (memory-like task), which involves a nonlinear relationship between RNN activations and behavior. We showed that while both b-dLDS and b-dLDS-x are able to reconstruct the behavior, b-dLDS can use a sparser set of components to do it.

4. Ablation and hyperparameter sensitivity experiments (on both simulated and real data): We tested the effects of Frobenius norm regularization and step size rescaling on the model’s performance with respect to ground truth dynamics reconstruction (2 independent systems simulation), Psi sparsity and structure (task-driven RNN), and neural and behavior data reconstruction (zebrafish); and the effects of model size, number of dynamics operators, and regularization on the sparsity of dynamics coefficients and behavior reconstruction (2 independent systems simulation and zebrafish).

---

### Decision · Action_Editor_sVsK · 2026-05-13

**Recommendation:** Accept with minor revision

**Additional Comments:**

After considering the reviewers’ comments and the authors’ responses, I recommend acceptance of the manuscript subject to the authors carefully addressing the remaining concerns in the final version. The paper presents a useful methodological contribution for neural data analysis and will likely be of interest to the AI4Science community. While some reviewers expressed reservations regarding the breadth of impact and positioning for a broader top-AI audience, the work nevertheless offers sufficient technical merit and relevance for publication.

At the same time, several important issues should still be addressed in the final manuscript. In particular, Reviewer 3 raised concerns regarding the derivation in Eqs. 15–21, noting that the current presentation appears mathematically inconsistent. The authors should carefully revisit this section, verify the correctness of the derivation, and provide clearer explanations and notation to eliminate any ambiguity or contradiction.

In addition, the manuscript would benefit from a clearer articulation of the novelty and broader significance of the proposed approach, particularly in relation to existing AI4Science methodologies. Strengthening the discussion on the potential implications and generalizability of the method would improve the paper’s positioning and accessibility to a wider AI audience.

Finally, the authors should ensure strict compliance with the official TMLR formatting guidelines. The current PDF appears to deviate from the standard Latin Modern font rendering expected from the official style files, raising concerns that the template may have been modified. The final version should therefore be carefully checked and regenerated using the official template without unintended modifications.

Subject to satisfactory revision of these points in the camera-ready version, I recommend the paper for publication.

**Audience:**

Yes

**Audience Explanation:**

Yes. The paper addresses a problem that is relevant to the AI4Science and computational neuroscience communities, and the proposed methodology for neural data analysis is likely to be of interest to researchers working at the intersection of machine learning and scientific applications. Although the scope and impact may be more specialized compared to broadly applicable AI methods, the work nevertheless provides useful technical insights and empirical findings that would be valuable to a subset of TMLR’s audience.

**Claims And Evidence:**

Yes

**Claims Explanation:**

The claims made in the submission are generally supported by empirical results and experimental evaluations, and the reviewers acknowledge that the proposed method is useful for neural data analysis and relevant to the AI4Science community. However, some concerns remain regarding the clarity and rigor of the theoretical derivation, particularly in Eqs. 15–21, where a reviewer identified a potentially unresolved mathematical inconsistency. While these concerns do not fundamentally invalidate the overall contribution, the theoretical presentation requires clarification and correction to ensure full technical soundness.

---

> ### Author Response · Authors · 2026-06-01
> **Thank you!**
>
> We would like to thank the AE for their recommendation of our paper for publication. We are working on addressing the requested edits and polishing our code for publication. We intend to have this done within the next couple of weeks. Thank you again!

---

> > ### Author Response · Authors · 2026-06-18
> > **Camera-ready version shared. Thank you!**
> >
> > Dear Action Editor and Reviewers,
> >
> > Thank you all for taking the time to review our paper. Your constructive comments have helped us improve the experimental validation and analyses into a more complete story, improve the clarity of the mathematical derivations, and position the paper more broadly within AI4Science.
> > To address the final specific updates, we 1) tidied up equations 15-21, 2) updated the Discussion to references other possible AI4Science applications of b-dLDS, 3) fixed the header of the document so as not to accidentally supersede the TMLR template’s font, and 4) included links to the public data and code repositories. Please see the updated camera-ready version.
> >
> > Thank you again,
> > the Authors